# Structure and distinct supramolecular organization of a PSII-ACPII dimer from a cryptophyte alga *Chroomonas placoidea*

Zhiyuan Mao[1,2,7], Xingyue Li[1,2,7], Zhenhua Li[1,2], Liangliang Shen [1,3,4], Xiaoyi Li[1,4], Yanyan Yang[1,4], Wenda Wang [1,4,5], Tingyun Kuang [1,4,5], Jian-Ren Shen [1,4,6] ✉ & Guangye Han [1,4,5] ✉

Cryptophyte algae are an evolutionarily distinct and ecologically important group of photosynthetic unicellular eukaryotes. Photosystem II (PSII) of cryptophyte algae associates with alloxanthin chlorophyll *a/c*-binding proteins (ACPs) to act as the peripheral light-harvesting system, whose supramolecular organization is unknown. Here, we purify the PSII-ACPII supercomplex from a cryptophyte alga *Chroomonas placoidea* (*C. placoidea*), and analyze its structure at a resolution of 2.47 Å using cryo-electron microscopy. This structure reveals a dimeric organization of PSII-ACPII containing two PSII core monomers flanked by six symmetrically arranged ACPII subunits. The PSII core is conserved whereas the organization of ACPII subunits exhibits a distinct pattern, different from those observed so far in PSII of other algae and higher plants. Furthermore, we find a Chl *a*-binding antenna subunit, CCPII-S, which mediates interaction of ACPII with the PSII core. These results provide a structural basis for the assembly of antennas within the supercomplex and possible excitation energy transfer pathways in cryptophyte algal PSII, shedding light on the diversity of supramolecular organization of photosynthetic machinery.

Oxygenic photosynthesis performed by cyanobacteria, various algae and higher plants are essential for survival of almost all life forms on the earth because it uses light energy to convert carbon dioxide and water into carbohydrates and oxygen, which provides the source of energy and molecular oxygen. The conversion of light energy into chemical energy occurs in two types of large pigment-protein complexes, photosystem I (PSI) and photosystem II (PSII), which are embedded in the thylakoid membranes of various oxygenic photosynthetic organisms[1]. Among them, PSII carries out light-induced electron transfer reactions coupled with the splitting of water and

release of protons and dioxygen[2-4]. The PSII supercomplex consists of a homodimeric reaction center core and peripheral antenna protein subunits[1,5], where the peripheral antennas absorb light energy and transfer them to the PSII core to initiate charge separation in the primary electron donor P680, a special pair of chlorophylls (Chls)[1-4].

In order to adapt to different light environments, different types of light-harvesting proteins (LHCs) and pigments (chlorophylls and carotenoids), as well as different supramolecular organizations between LHCs and PSII cores, have been evolved in various types of photosynthetic organisms during evolution[5-7]. Among them, the

[1]Photosynthesis Research Center, Key Laboratory of Photobiology, Institute of Botany, Chinese Academy of Sciences, 100093 Beijing, China. [2]University of Chinese Academy of Sciences, 100049 Beijing, China. [3]Cryo-EM Centre, Southern University of Science and Technology, 518055 Guangdong, China. [4]China National Botanical Garden, 100093 Beijing, China. [5]Academician Workstation of Agricultural High-tech Industrial Area of the Yellow River Delta, National Center of Technology Innovation for Comprehensive Utilization of Saline-Alkali Land, 257300 Dongying, China. [6]Institute for Interdisciplinary Science, and Graduate School of Natural Science and Technology, Okayama University, Okayama 700-8530, Japan. [7]These authors contributed equally: Zhiyuan Mao, Xingyue Li. ✉e-mail: shen@cc.okayama-u.ac.jp; hanguangye@ibcas.ac.cn

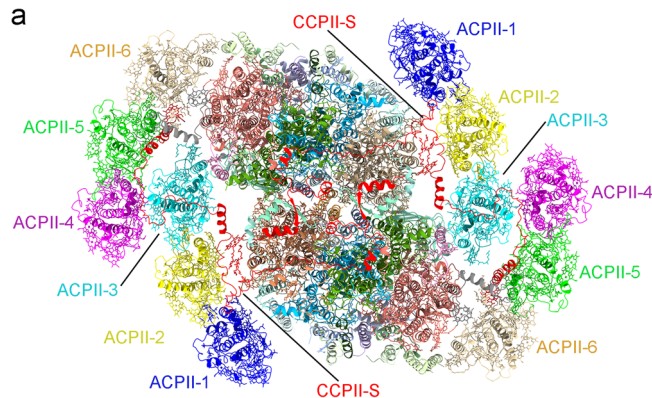

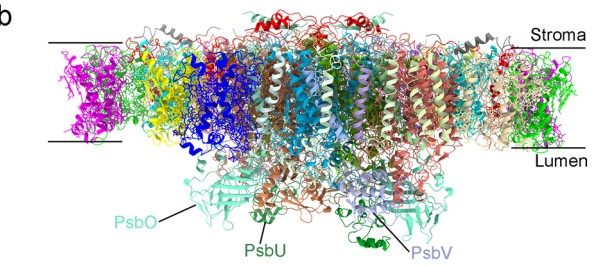

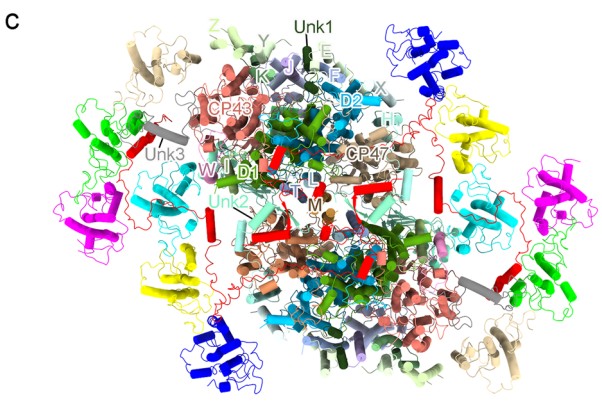

**Fig. 1 | Overall structure of the PSII-ACPII supercomplex of *C. placoidea*.**
**a** Overall structure of the PSII-ACPII supercomplex viewed normal to the membrane plane from the stromal side. All ACPII subunits and the CCPII-S subunit are labeled.
**b** Side view of the PSII-ACPII supercomplex with the three extrinsic proteins labeled. **c** Top view of the PSII-ACPII supercomplex from the stromal side with the core subunits and three unknown subunits (Unk1, Unk2, and Unk3) labeled.

structure of the PSII core is relatively conserved from cyanobacteria to higher plants. On the other hand, water-soluble phycobilisome proteins (PBSs) are associated with the stromal side of the cyanobacterial and red algal PSII cores as the light-energy harvesting system[8–12], whereas variable types and numbers of Chl *a/b* or Chl *a/c* binding trans-membrane LHC proteins are associated with the green algal, diatom and higher plant PSII cores to serve as the peripheral antenna systems[13–18].

Cryptophyte algae are a lineage of unicellular eukaryotic phytoplankton that live in marine, brackish water, and freshwater, and play a crucial ecological role in the global carbon and biogeochemical cycles[19,20]. Cryptophyte algae acquired their photosynthetic plastids through secondary endosymbiosis between an unknown eukaryotic host and a red algal ancestor during evolution[20–24]. Thus, similar to the red algae, cryptophyte algae represent one of the groups of photosynthetic organisms that use a specific light-harvesting antenna system composed of water-soluble phycobiliproteins and membrane intrinsic

Chl antenna proteins[25]. However, the location, structure and type of phycoerythrin/phycocyanin present in phycobiliproteins of cryptophyte algae differ from phycobilisomes of red algae and cyanobacteria[26,27], and the cryptophytic equivalent of Chl antenna are membrane intrinsic, three-α-helix proteins, evolutionarily closely related to the light-harvesting complexes (LHCs) of red algae[28,29].

The pigment-protein complexes of PSI and PSII of cryptophytes are localized in thylakoid membranes as monomers for PSI and as dimers for PSII, respectively, similar to those seen in other algae and higher plants[30,31]. The structure of PSI-light-harvesting antenna (LHCI) supercomplex from a cryptophyte alga *Chroomonas placoidea* (*C. placoidea*) has been determined by cryo-electron microscopy recently and reveals detailed composition and structural features of the PSI-LHCI complex[32]. On the other hand, the structure of PSII-LHCII from cryptophyte algae has not been determined. Previously, a two-dimensional projection map related to PSII supercomplex from a cryptophyte *Rhodomonas* CS24 has been obtained, showing a dimeric organization of the PSII core with monomeric Chl antenna proteins associated preferentially in one side of the PSII core[30]. However, detailed knowledge regarding the number and structure of antenna protein subunits and pigment composition, the assembly pattern of antennas to the PSII core, and the energy transfer pathways within this supercomplex are elusive because of the absence of a high-resolution structure of the complete PSII supercomplex.

In this work, we isolate the PSII-ACPII supercomplex from a cryptophyte alga *C. placoidea* and solve its structure using single-particle cryo-electron microscopy (cryo-EM). Three-dimensional cryo-EM density map of a PSII-ACPII dimer is obtained at an overall resolution of 2.47 Å. Our structure reveals a distinct organization of PSII supercomplex in *C. placoidea* and provides insights into the protein structures and pigment arrangement, as well as energy transfer pathways within this large complex, thus providing a structural basis to understand the molecular assembly and energy transfer mechanism from the peripheral antenna to the PSII core.

## Results
### Overall structure of the PSII-ACPII supercomplex
The PSII-ACPII supercomplex was isolated from the cryptophyte alga *C. placoidea* and characterized using SDS-polyacrylamide gel electrophoresis (SDS-PAGE), size-exclusion chromatography, absorption and fluorescence spectroscopy, and HPLC (Materials and Methods, Supplementary Fig. 1). The results showed that the purified sample consisted of major components of the PSII core and a large quantity of ACPII subunits, and was homogenous enough in size to be used for cryo-EM analysis. We collected a total of 10,663 cryo-EM movies and selected 1,263,069 particles for subsequent data processing. After 2D classification and refinement, particles in the majority group were selected for further data processing, which resulted in a cryo-EM map of the PSII-ACPII supercomplex with an overall resolution of 2.47 Å (Supplementary Fig. 2 and Table 1). Targeted local refinement was conducted to increase the resolution of the cryo-EM density maps for the peripheral ACPIIs, leading to a final local resolution of 2.84 Å for the peripheral antennas of PSII-ACPII (Supplementary Fig. 2).

The overall structure of the PSII-ACPII supercomplex is a homodimer with a *C2* symmetry, and is composed of two PSII core monomers and twelve ACPII subunits, with each PSII core monomer binding six ACPII subunits (Fig. 1 and Supplementary Fig. 3). In addition, a special Chl *a*-binding antenna subunit was identified and named as cryptophyte Chl *a*-binding special protein (CCPII-S) in each PSII-ACPII monomer (Fig. 1a). In addition to protein subunits, we identified a variety of pigments and lipids within the PSII-ACPII supercomplex, which include 208 Chl *a*, 12 Chl *c*, 42 alloxanthin (Alx), 24 α-carotene (α-Car), 10 crocoxanthin (Cro), and 4 monadoxanthin (Mon) molecules as well as 46 lipids including 20 edistearoylmonogalactosyl diglyceride (LMG), 8 sulfoquinovosyldiacylglycerol (SQD), and 18

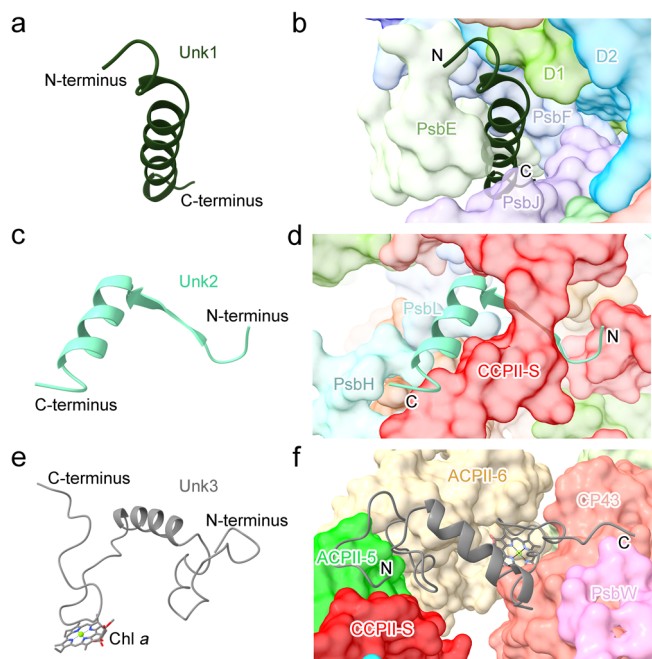

**Fig. 2 | Structures and locations of the Unk1, Unk2, and Unk3 subunits in PSII-ACPII of *C. placoidea*. a, b** The structure of Unk1 (**a**) and its location in the supercomplex (**b**). **c, d** The structure of Unk2 (**c**) and its location in the supercomplex (**d**). **e, f** The structure of Unk3 (**e**) and its location in the supercomplex (**f**). The adjacent subunits of Unk1, Unk2, and Unk3 are shown in surface mode.

dipalmitoylphosphatidyl glycerol (LHG) molecules (Supplementary Fig. 3 and Supplementary Table 2 and Table 3).

## Structure of the *C. placoidea* PSII core

The PSII core is composed of four large transmembrane subunits (D1, D2, CP47, and CP43), thirteen small, transmembrane subunits (PsbE, PsbF, PsbH, PsbI, PsbJ, PsbK, PsbL, PsbM, PsbT, PsbW, PsbX, PsbY, and PsbZ) (Fig. 1c), and three lumenal extrinsic subunits (PsbO, PsbU, PsbV) involved in stabilization of the oxygen-evolving complex (Fig. 1b). These protein subunits are similar to the cyanobacterial PSII core whose structure has been analyzed[3,4,33]. In addition, three transmembrane helices were found in the PSII core (Fig.1c), but their amino acid residues could not be identified due to the limited resolution. These subunits were hence modeled as poly-alanines and named unknown proteins Unk1, Unk2, and Unk3, respectively (Fig. 1c). Structural comparison of the PSII cores reveals that the root-mean-square deviation (RMSD) values of α-carbon atoms between the PSII core of cryptophyte algae and that of a cyanobacterium[4], a red alga[34], a green alga[14], a diatom[16], and a higher plant[18] (including only the same subunits in these PSII cores) are 0.827, 0.910, 1.035, 0.743, and 0.983, respectively (Supplementary Fig. 4a–e). This suggests that the PSII core subunits of the cryptophyte alga share high similarities with those from other organisms.

All of the three unknown proteins have a single transmembrane helix. Unk1 is located proximal to the subunits PsbJ, PsbF, and PsbE (Fig. 1c and Fig. 2a, b). No subunit is observed in the corresponding locations in the PSII core of cyanobacteria, red algae, green algae, diatoms, and higher plants. Unk2 is located at the interface of two PSII monomers and is in close proximity to the C-terminal region of the herein identified CCPII-S subunit (Fig. 1a, c and Fig. 2c, d). Unk3 associates with CP43 on the stromal side of the PSII-ACPII supercomplex and interacts with the nearby ACPII-5 subunit, facilitating connections of the ACPII antenna subunit with the PSII core (Fig. 1a, c and Fig. 2e, f). Different from Unk1 and Unk2, Unk3 binds a Chl *a* molecule in a loop region that locates at the gap between ACPII-6 and

CP43, contributing to energy transfer from ACPII to the PSII core (Fig. 2e, f).

## Structure and organization of the peripheral antennas

Six ACPII antenna are bound to each PSII monomer. It has been shown that cryptophyte algae contain two LHC gene families, *lhcr* and *lhcz*[29], which are also detected in our transcriptome analysis. The high-quality cryo-EM density and sequence characteristics of the ACPII protein isoforms allow us to model the six ACPIIs as antennas encoded by the *lhcr* genes rather than by the *lhcz* genes in the structure (Supplementary Fig. 5), and they are named as ACPII-1/2/3/4/5/6. Among them, ACPII-1/2/3 are associated at the PsbX-PsbH-CP47 side, and ACPII-4/5/6 are associated at the PsbW-CP43-PsbZ side, respectively (Fig. 1). Thus, three ACPII subunits forms an antenna belt in each side of the PSII core monomer, but their structures are different from the trimeric or tetrameric antenna forms observed in green algae, diatoms, or higher plants[13–18]. All ACPII antenna subunits consist of three transmembrane helices (αA, αB, αC), a short amphipathic helix αE between αA and αB, and a short amphipathic helix αD at their C-terminal ends (Supplementary Fig. 6a), which are similar to those of LHCs from red algae, green algae, diatoms and higher plants[13–18]. However, structural comparison of the six ACPII subunits reveals remarkable differences in the lengths and orientations of the N and C terminuses. At the N-terminus, ACPII-1 and ACPII-5 have the longest N-terminal tails but with opposite orientations, whereas both ACPII-4 and ACPII-6 have the shortest N-terminal tails. The N-terminal length of ACPII-2 is similar to that of ACPII-3. At the C-terminus, the C-terminal tails of ACPII-1 and ACPII-5 are the longest in length and have opposite orientations, whereas the C-terminal length of ACPII-4 is the shortest. ACPII-2, ACPII-3, and ACPII-6 have similar C-terminal lengths (Supplementary Fig. 6a). These structural differences may be a result of the specific position that each antenna subunit occupies, thus contributing to the distinct arrangement and stable assembly of the ACPIIs in the PSII-ACPII supercomplex.

In the twelve ACPII subunits of the whole dimer supercomplex, 132 Chl *a*, 12 Chl *c*, 42 Alx, 10 Cro, 4 Mon, and 4 α-Car molecules are identified (Supplementary Table 2 and Table 3). Each ACPII subunit binds 9 to 12 Chls *a*, and most of the binding sites of Chl *a* are conserved across the six ACPII subunits, with the exception of *a*613 and *a*614 specific to ACPII-1 and ACPII-4, respectively, and *a*606 missing in ACPII-3/6. In addition, ACPII-1/2/4/5/6 each also binds 1–2 Chls *c*, whereas ACPII-3 does not bind any Chl *c* and a Chl *a* molecule is found in the binding site corresponding to *c*610 (Fig. 3, Supplementary Fig. 6b, c and Supplementary Table 3). In addition to the large amounts of Chls, each ACPII subunit binds 3–4 Alx molecules. The subunits of ACPII-1/2/4/5 each binds 1–2 Cros, and each subunit of ACPII-1-6 binds a single Mon, whereas both ACPII-3 and ACPII-6 bind one α-carotene (Fig. 3, Supplementary Fig. 6b, c and Supplementary Table 3).

The central Mg atoms of Chls of ACPIIs are coordinated by side chains of different amino acid residues. Chls *a*602, *a*603, *a*604, *a*605 (or *c*605), *a*606, *a*607, *a*608, and *a*612 are coordinated by Glu, His, Gln, Gln, His, Glu, Glu, and His, respectively. The axial ligand of *a*601 is Ile in ACPII-1, Ala in ACPII-2/3/4 /6, and Ser in ACPII-5, respectively. A lipid molecule LHG621 is specifically identified as the axial ligand of *a*609 in ACPII-2, whereas its counterpart in ACPII-1/3/4/5/6 is difficult to discern due to weak densities. Chl *c*610 is ligated by Asn in ACPII-1/4/6 and His in ACPII-2/5, whereas Chl *a*610 in ACPII-3 is connected to a His residue. Chl *a*611 is coordinated by His in ACPII-1/4, and Gln in ACPII-2/3/5/6. Notably, *a*613, exclusive to ACPII-1, lacks a discrete ligand, and *a*614 in ACPII-4 is coordinated by Pro (Supplementary Fig. 7).

CCPII-S is a distinct antenna subunit consisting of a single N-terminal transmembrane helix and an extended C-terminus, with a total molecular weight of around 24 kDa. The N-terminal transmembrane helix is located at the junction of ACPII-5, ACPII-6, and CP43, whereas the extended C-terminus is composed of six α-helices and six

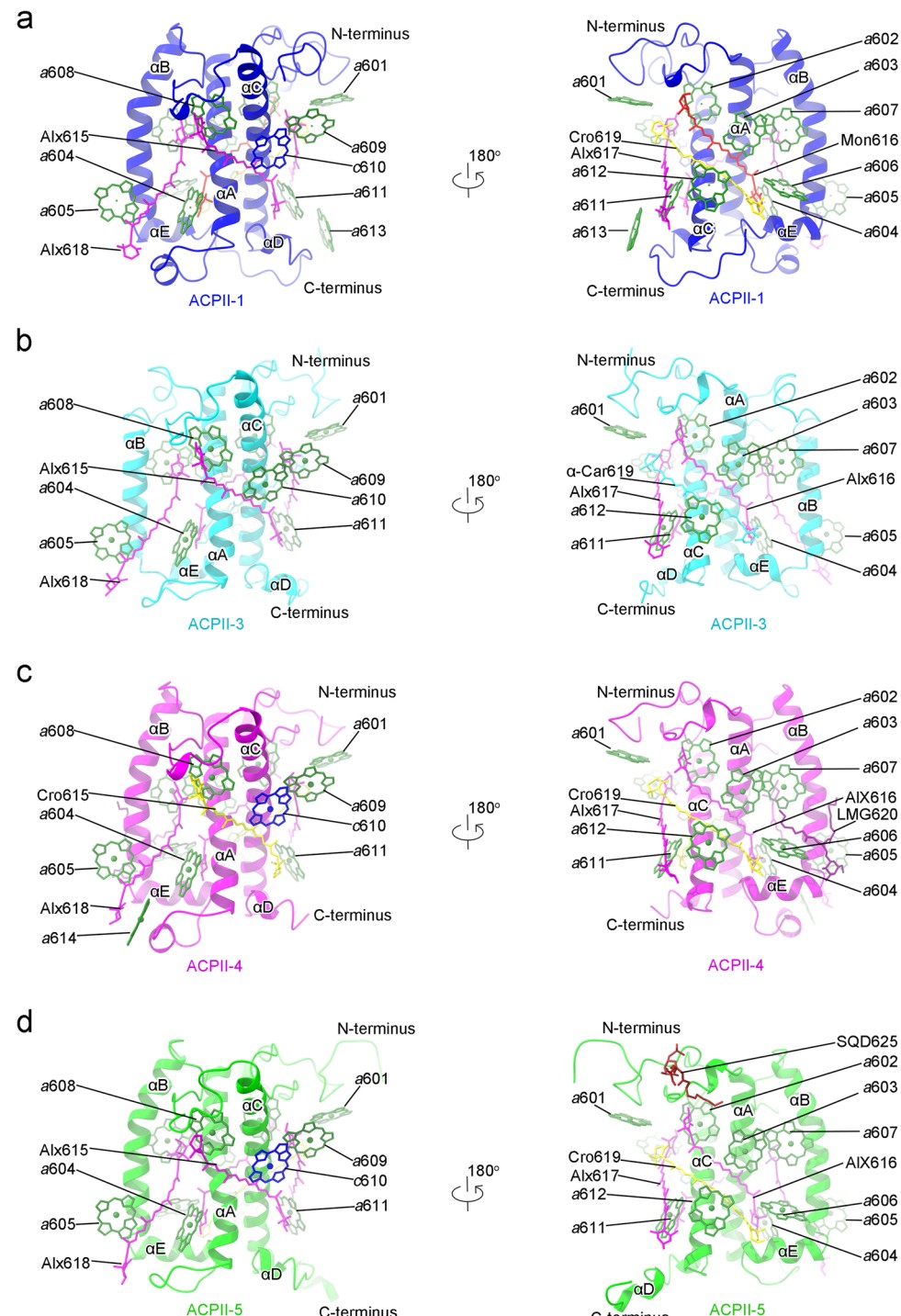

**Fig. 3 | Structures of individual ACPII subunit of *C. placoidea* PSII-ACPII.**
**a–d** Structures of ACPII-1 (**a**), ACPII-3 (**b**), ACPII-4 (**c**) and ACPII-5 (**d**) and arrangements of pigments and lipids. The apo-protein structures of ACPII subunits are depicted in transparent cartoon mode, whereas Chl *a*, Chl *c*, alloxanthin, α-carotene, monadoxanthin, crocoxanthin, LMG and SQD are depicted as sticks and colored as green, blue, magenta, cyan, red, yellow, purple and brown, respectively. For clarity, the phytol chain of the Chl molecules are omitted.

loops located at the stromal side, and is involved in the interactions with ACPII subunits and PSII core subunits to mediate the association of ACPII subunits with the PSII core (Fig. 4a, b). In addition, two Chl *a* molecules, Chl *a*601 and Chl *a*602, are identified in CCPII-S (Fig. 4c, d). Chl *a*601 is located in the transmembrane helix region and associates with CCPII-S through a hydrogen-bond interaction with His93 at a distance of 3.4 Å (Fig. 4e), whereas Chl *a*602 binds to CCPII-S via ligation of Mg by Trp163 and a hydrogen-bond with Leu168 at a distance of 3.3 Å (Fig. 4f).

Structural comparison of ACPII and ACPI antennas from PSII-ACPII and PSI-ACPI[32] reveals structural similarities between them (Supplementary Fig. 8). The three typical transmembrane helices (αA, αB, αC) in these antennas are almost completely overlapped. Notably, the amino acid sequences of ACPII-2 and ACPII-6 are identical with that of ACPI-2 and ACPI-14, respectively, and the protein structures, pigments bound are exactly the same between ACPII-2 and ACPI-2 as well as ACPII-6 and ACPI-14 (Supplementary Fig. 8c, g).This implies that ACPII-2 and ACPII-6 subunits are encoded by the same gene as that of ACPI-2

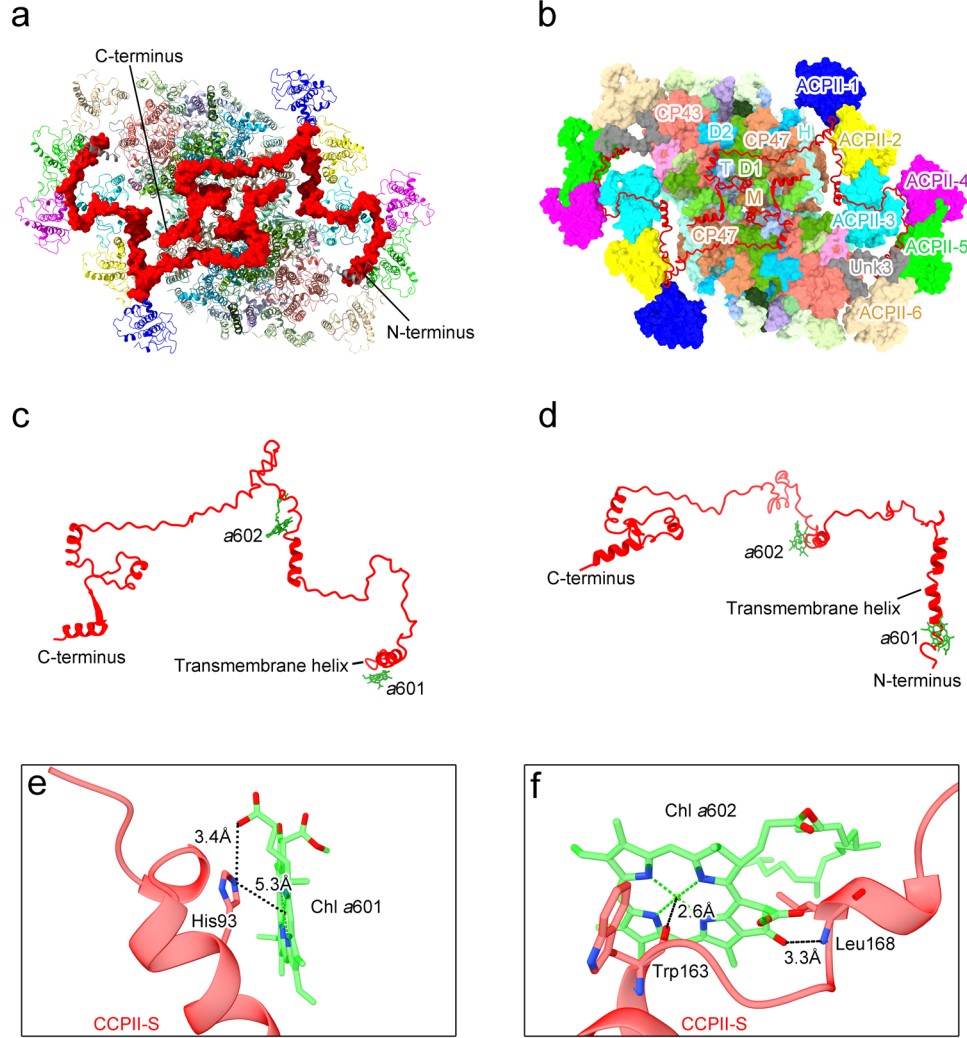

**Fig. 4 | Location and structure of the CCPII-S subunit in the PSII-ACPII supercomplex of *C. placoidea*. a** Surface representation of the location of CCPII-S within the PSII-ACPII supercomplex viewed from the stromal side. **b** Subunits involved in the interactions with CCPII-S in the PSII-ACPII at the stromal side. The subunits involved in the interactions with CCPII-S are labeled. **c, d** The structure of CCPII-S viewed from the top (**c**) and side (**d**), respectively. **e, f** Interactions of CCPII-S amino acid residues with the bound pigments Chl *a*601 (**e**) and Chl *a*602 (**f**), respectively.

and ACPI-14, respectively, and ACPII-2 and ACPII-6 (or ACPI-2 and ACPI-14) can exist and function in both PSI and PSII simultaneously, which has not yet been observed in other species. In addition, superposition on the basis of the ACPII-2 and ACPI-2 subunits revealed a high overall structural similarity between ACPII-1/2/3 trimer of PSII-ACPII and ACPI-1/2/3 trimer of PSI-ACPI[32] (Supplementary Fig. 9a). Similarly, upon superposition on the basis of the ACPI-6 and ACPI-14 subunits, ACPII-4/5/6 trimer of PSII-ACPII showed a high overall structural similarity with the ACPI-12/13/14 trimer of PSI-ACPI[32] (Supplementary Fig. 9b). Moreover, a similar case was observed when structural comparison of the moiety of ACPII-1/2/3/4/5/6 in PSII-ACPII with that of ACPI-1/2/3/12/13/14 in PSI-ACPI was performed with either ACPII-2/ACPI-2 or ACPII-6/ACPI-14 as the basis for superposition (Supplementary Fig. 9c, d). The similarities of the antenna structure and organization between PSI and PSII suggests that both photosystems use some very similar or even common set of the antenna subunits, which may suggest that these antenna subunits have not been diversified as seen in diatoms or green algae.

### Assembly of the PSII-ACPII supercomplex

For ACPII-1/2/3 or ACPII-4/5/6 bound to the same side of PSII, direct hydrophobic interactions between Chl *a*605 of ACPII-n and Chl *a*611/Alx617 of ACPII-(n + 1) are found, which may contribute to the associations of adjacent ACPII subunits (Fig. 5a–d). In addition, hydrogen-bond interactions between $E149_{ACPII-2}$ and $S54_{ACPII-3}$, $Q159_{ACPII-4}$ and $K45_{ACPII-5}$, as well as $K146_{ACPII-5}$ and $E53_{ACPII-6}$ are found, which may benefit the direct association of adjacent ACPII subunits (Fig. 5e–g). For ACPII-3 from one PSII monomer and ACPII-4 from the adjacent PSII monomer, the association between them is mediated by hydrophobic interactions between Chl $a609_{ACPII-3}$ and Chl $a603_{ACPII-4}$ as well as Chl $a610_{ACPII-3}$ and Chl $a607_{ACPII-4}$ (Fig. 5h).

Regarding the connections between the ACPII antenna subunits and the PSII core, only a few direct associations are found, due to the relatively long distances between ACPII-1/2/4/5/6 and the PSII core. ACPII-3 contacts directly to the PSII core by hydrophobic interactions formed between Chl *a*607 bound in the αB region and residues of the PsbW subunit (Fig. 5i). Additionally, ACPII-5 contacts with Unk3 which binds to the core of another PSII monomer by hydrogen-bond interactions of Unk3 residues with K243, N143, and T147 of CP43, respectively (Fig. 5j), contributing to the connections of the antenna subunit with the PSII core. Unk3 is positioned at the interface of ACPII-5 and ACPII-6, and the A6 residue of Unk3 forms a hydrogen-bond with Q144 of ACPII-5 (Fig. 5g). These interactions may enhance the stable binding of the ACPII subunits to the PSII core in cryptophyte algae.

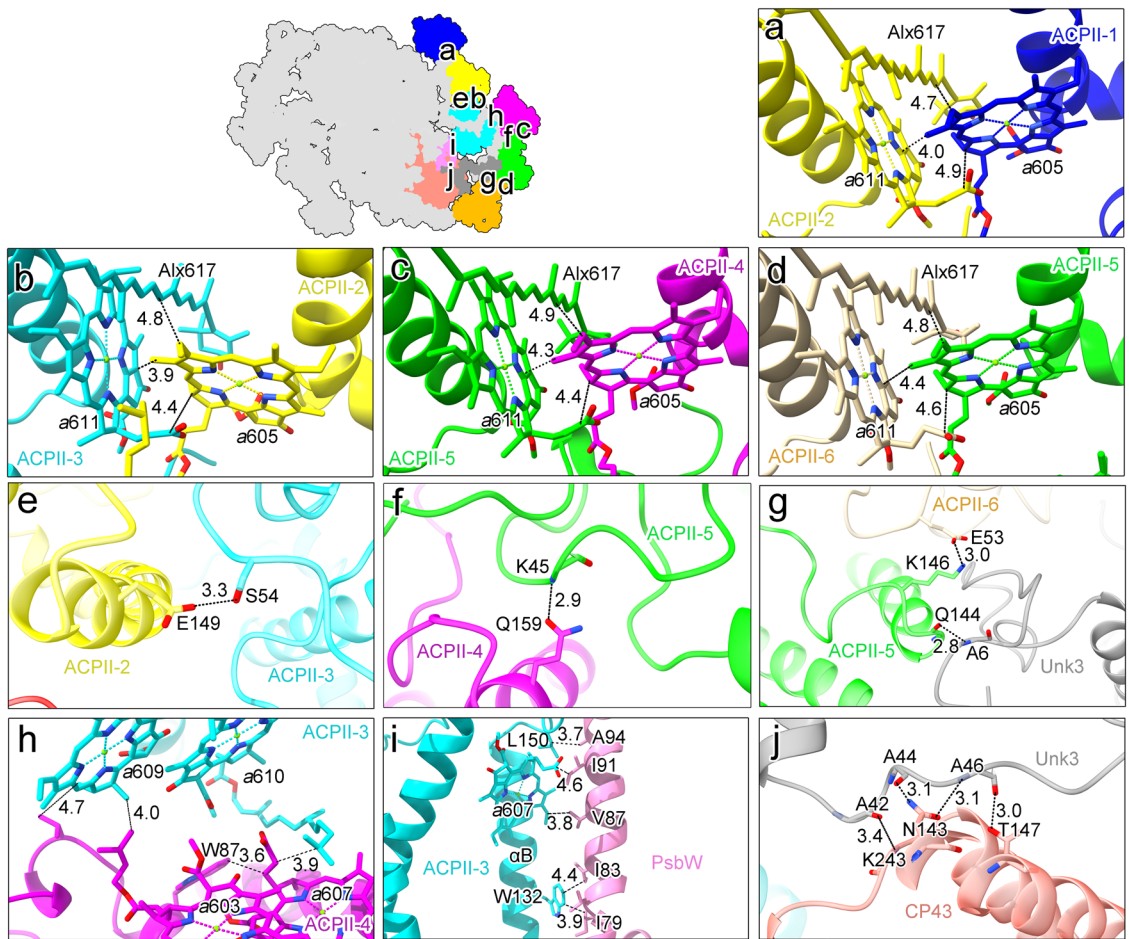

**Fig. 5 | Antenna-antenna and antenna-core interactions in the PSII-ACPII supercomplex of *C. placoidea*. a–d** Hydrophobic interactions formed between ACPII-1 and ACPII-2 (**a**), ACPII-2 and ACPII-3 (**b**), ACPII-4 and ACPII-5 (**c**), and ACPII-5 and ACPII-6 (**d**). **e–g** Hydrogen-bond interactions formed between ACPII-2 and ACPII-3 (**e**), ACPII-4 and ACPII-5 (**f**), and ACPII-5 and ACPII-6 as well as ACPII-5 and Unk3 (**g**). **h** Hydrophobic interactions formed between ACPII-3 and ACPII-4. **i** Hydrophobic interactions formed between ACPII-3 and the PSII core subunit PsbW. **j** Hydrogen-bond interactions formed between Unk3 and the PSII core subunit CP43.

In addition to the above-mentioned interactions, the association between ACPII and the PSII core is facilitated by the special Chl *a* binding antenna subunit, CCPII-S, which interacts with the ACPII subunits and PSII core subunits extensively, therefore acting as a bridge mediating the connections between ACPII and the PSII core. The N-terminal domain of CCPII-S is tightly connected with the ACPII subunits, while its C-terminal domain pervades the entire PSII core. The N-terminal transmembrane helix of CCPII-S is located in a cavity surrounded by ACPII-5, ACPII-6, and CP43, and is associated with ACPII-5 by hydrophobic interactions between $V97_{CCPII-S}$ and $L125_{ACPII-5}$ and between $V108_{CCPII-S}$ and $F140_{ACPII-5}$ (Fig. 6a). On the other hand, the C-terminus of CCPII-S extends to the interface of ACPII-4, ACPII-3, ACPII-2 and ACPII-1 along the stromal side, and after a 90° turn, it sequentially stretches across the PSII core subunits PsbH, CP47, D1, D2, CP43, PsbT, PsbM at the stromal side, and finally ends with two hydrophilic helices involved in the interactions with CP47 of another PSII monomer at the stroma side. At the interface between ACPII-3 and ACPII-4, CCPII-S contacts with ACPII-4 by a hydrogen-bond formed between $T130_{CCPII-S}$ and $V82_{ACPII-4}$ (Fig. 6b). Subsequently, the C-terminus of CCPII-S undergoes a 90° turn on the ACPII-3 side and contacts with ACPII-3 by multiple hydrogen-bond interactions formed between $P144_{CCPII-S}$ and $K81_{ACPII-3}$, $K141_{CCPII-S}$ and $D164_{ACPII-3}$, as well as between $L138_{CCPII-S}$ and $S62_{ACPII-3}$ (Fig. 6c). At the intersection between ACPII-1 and ACPII-2, extensive hydrogen-bond interactions are formed between $P184_{CCPII-S}$ and $N163_{ACPII-1}$,

$D187_{CCPII-S}$ and $Q160_{ACPII-1}$, $R186_{CCPII-S}$ and $D44_{ACPII-2}$, $T188_{CCPII-S}$ and $L71_{ACPII-2}$, $R174_{CCPII-S}$ and $V76_{ACPII-2}$ (Fig. 6d). Thus, these CCPII-S mediated interactions enhance the binding stability of ACPII antennas.

Extensive interactions are also observed between CCPII-S and the PSII core subunits. CCPII-S contacts with PsbH in two sites. While $E156_{CCPII-S}$ forms a hydrogen-bond with $R6_{PsbH}$ in the N-terminus (Fig. 6e), $D208_{CCPII-S}$ is involved in hydrogen-bond interactions with $G19_{PsbH}$ (Fig. 6f). CCPII-S is hydrogen-bounded with CP47 between $L216_{CCPII-S}$ and $D477_{CP47}$, $E210_{CCPII-S}$ and $R230_{CP47}$, $Q212_{CCPII-S}$ and $Y226_{CP47}$ (Fig. 6f). S250 and E253 of CCPII-S form hydrogen-bonds with E231 and K238 of D1 at distances of 2.5 and 3.0 Å, respectively (Fig. 6g). Finally, the C-terminus of CCPII-S spans the interface of two PSII core monomers and forms hydrogen-bond interactions with PsbL and CP47 of the adjacent PSII core monomer. $R279_{CCPII-S}$ and $K285_{CCPII-S}$ form hydrogen-bonds with $P4_{PsbL}$ and $P126_{CP47}$ of the adjacent PSII monomer, respectively (Fig. 6h). These interactions are specific to PSII-ACPII of *C. placoidea* and contribute to the stability of the PSII dimer in the cryptophyte algae.

### Possible energy transfer pathways in the PSII-ACPII supercomplex
Based on the high resolution density map, 300 pigment molecules are identified in the PSII-ACPII supercomplex of cryptophyte alga *C.placoidea* (Fig. 7a and Supplementary Tables 2, 3). Chls are

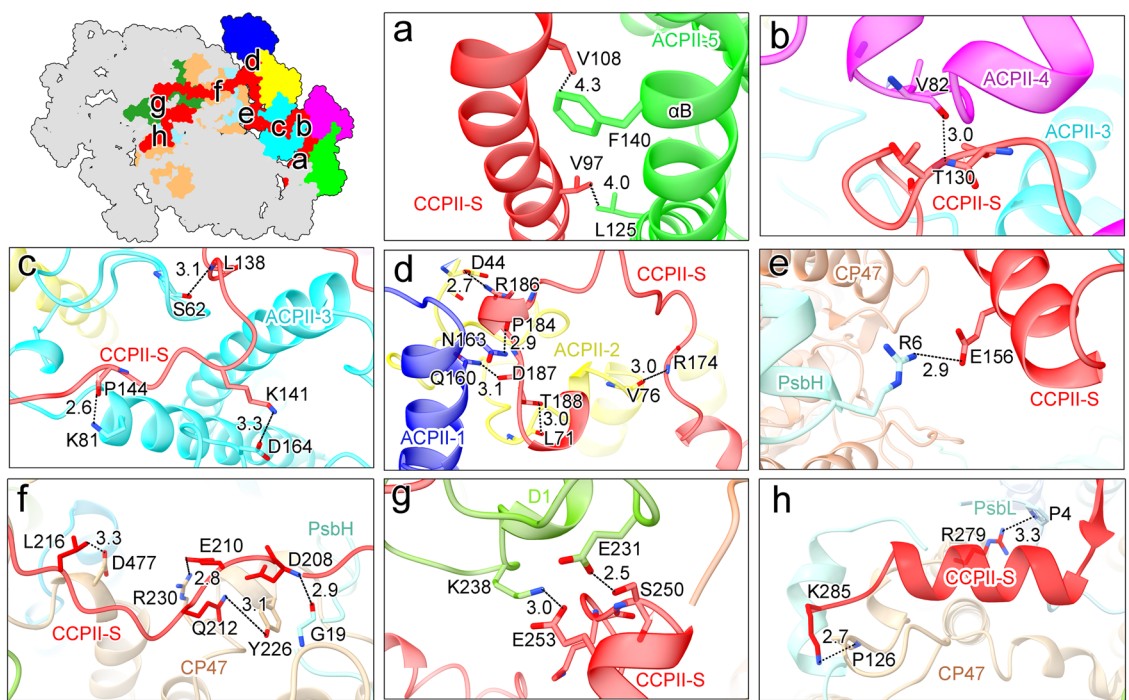

**Fig. 6 | Interactions of CCPII-S with both ACPIIs and the PSII core in the PSII-ACPII supercomplex of *C. placoidea*. a** Hydrophobic interactions between CCPII-S and ACPII-5. **b–d** Hydrogen-bond interactions of CCPII-S with ACPIIs subunits, specifically with ACPII-4 (**b**), ACPII-3 (**c**), ACPII-2 and ACPII-1 (**d**); **e–h** Hydrogen-bond interactions of CCPII-S with PSII core subunits, including PsbH (**e**), CP47 and PsbH (**f**), D1 (**g**), CP47 and PsbL from another PSII monomer (**h**).

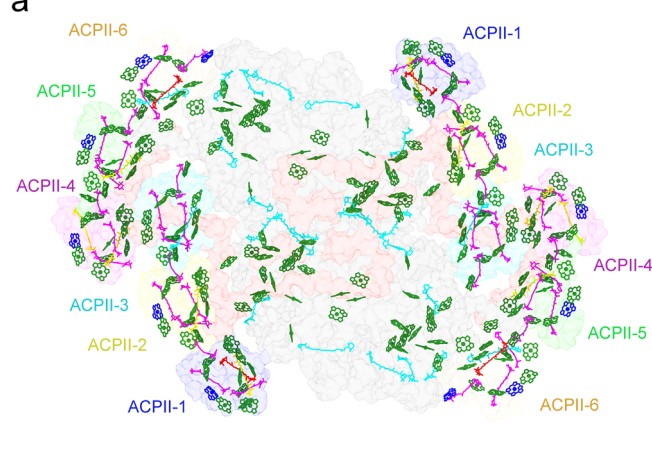

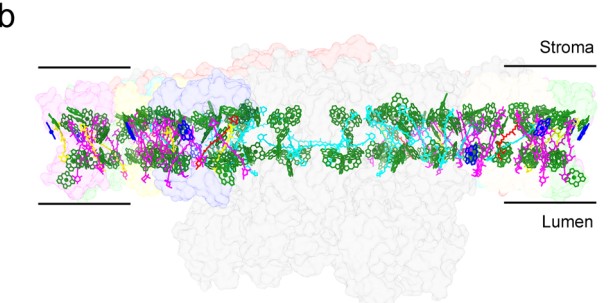

**Fig. 7 | Overall distributions of pigments in the PSII-ACPII supercomplex of *C. placoidea*. a, b** Distributions of pigments (Chls and carotenoids) in PSII-ACPII viewed normal to the membrane plane from the stromal side (**a**) and the side view along the membrane plane (**b**). Chl *a*, Chl *c*, alloxanthin, α-carotene, monadox-anthin and crocoxanthin are colored as green, blue, magenta, cyan, red and yellow, respectively. For clarity, the phytol chains of the Chl molecules have been omitted.

categorized into two layers, namely, the stromal-side layer and the lumenal-side layer (Fig. 7b). The number of Chls in these two layers is unequal, with 62 and 48 Chls in each PSII monomer close to the stromal and lumenal sides, respectively. All Chls *c* are located in the peripheral regions of the ACPII subunits, with 5 Chl *c* on the stromal side and 1 Chl *c* on the lumenal side layer in each monomer. The 5 stromal side Chls *c* have short distances with the nearby Chls *a*, suggesting fast energy coupling of Chl *c* with Ch *a*. In the ACPII antennas, all Cars are located in positions close to Chls, enabling fast the energy transfer and quenching between them. Possible pathways for excitation energy transfer (EET) between ACPII antennas and from ACPII antennas to the PSII core are proposed based on the Förster resonance energy transfer (FRET) network calculated from the distances and orientations of pigments resolved in the present study (Fig. 8). For the FRET calculations, we estimated the energy transfer rate between Chl *a* molecules with a Mg-Mg distance of less than 30 Å within the PSII-ACPII supercomplex (Fig. 8)

Regarding the energy transfer among the ACPII subunits, the pigment pairs between Chl $a608_{ACPII-1}$ and Chl $a601_{ACPII-2}$, between Chl $a608_{ACPII-2}$ and Chl$a$ $601_{ACPII-3}$, between Chl $a608_{ACPII-4}$ and Chl $a601_{ACPII-5}$, and between Chl $a608_{ACPII-5}$ and Chl $a601_{ACPII-6}$, have a FRET rate of 0.775, 0.792, 0.589 and 0.765 ps$^{-1}$, respectively (Supplementary Table 4). Thus, efficient EET may occur between the two closest pigments, Chl $a601$ and Chl $a608$ in the two adjacent ACPII subunits at the stromal side (Fig. 9a). In the lumenal side, FRET rates between Chl $a605_{ACPII-1}$and Chl $a611_{ACPII-2}$, between Chl $a605_{ACPII-1}$ and Chl $a612_{ACPII-2}$, between Chl $a605_{ACPII-2}$ and Chl $a611_{ACPII-3}$, and between Chl $a605_{ACPII-2}$ and Chl $a612_{ACPII-3}$, are 0.962, 0.509, 1.321, and 0.383 ps$^{-1}$, respectively, which suggest these three closest Chls *a* ($a605$, $a611$ and $a612$) may mediate EET between the two adjacent ACPII subunits of ACPII-1, ACPII-2, and ACPII-3 (Fig. 9a and Supplementary Table 4). Similar EET pathways mediated by three closest Chls *a* were observed between two adjacent ACPII subunits of ACPII-4, ACPII-5, and ACPII-6 (Fig. 9a and Supplementary Table 4). In addition,

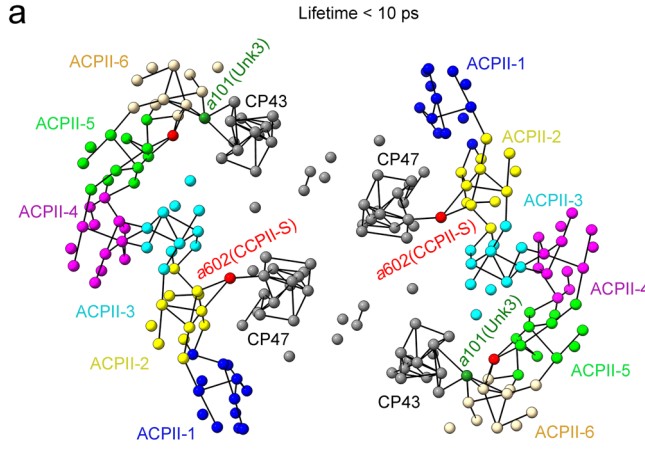

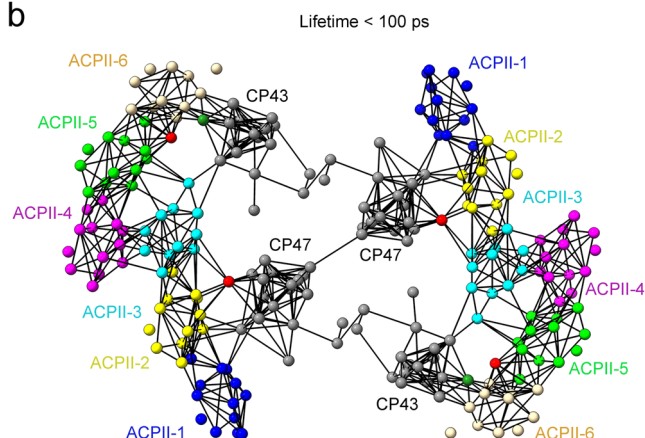

**Fig. 8 | Structure-based calculation of FRET rates presenting possible excitation energy transfer pathways in the PSII-ACPII supercomplex.**
**a**, **b** Organization of Chl molecules and the FRET networks presenting efficient FRET processes with lifetimes of less than 10 ps (**a**) and 100 ps (**b**) in the PSII-ACPII supercomplex viewed from the stromal side. Chl molecules are shown as spheres and FRET processes between adjacent Chls are shown as lines.

the Chl pairs between Chl $a606_{ACPII-5}$ and Chl $a601_{CCPII-S}$, and between Chl $a601_{CCPII-S}$ and Chl $a612_{ACPII-6}$, have a FRET rate of 0.350 ps$^{-1}$ and 0.835 ps$^{-1}$, suggesting Chl $a601$ from CCPII-S may mediate the energy transfer between Chl $a606_{ACPII-5}$ and Chl $a612_{ACPII-6}$ in the lumenal side (Fig. 9a and Supplementary Table 4). For the EET from the outer antenna ACPII-4 to the inner antenna ACPII-3, the FRET rates between Chl $a603_{ACPII-4}$ and Chl $a610_{ACPII-3}$ with a Mg-Mg distance of 13.70 Å, and between Chl $a603_{ACPII-4}$ and Chl $a609_{ACPII-3}$ with a Mg-Mg distance of 13.96 Å are 0.163 ps$^{-1}$ and 0.603 ps$^{-1}$, indicating that efficient EET may take place from Chl $a603_{ACPII-4}$ to Chl $a610_{ACPII-3}$ and Chl $a609_{ACPII-3}$ at the stromal side (Fig. 9a, b and Supplementary Table 4). The FRET rate between Chl $a607_{ACPII-4}$ and Chl $a610_{ACPII-3}$ with a Mg-Mg distance of 13.74 Å is 0.734 ps$^{-1}$, suggesting that this Chl pair may contribute to the efficient energy transfer at the stromal side (Fig. 9a, b and Supplementary Table 4). In addition, low FRET rates between Chl $a606_{ACPII-4}$ and Chl $a611_{ACPII-3}$, between Chl $a606_{ACPII-4}$ and Chl $a604_{ACPII-3}$, between Chl $a612_{ACPII-5}$ and Chl $a604_{ACPII-3}$, and between Chl $a606_{ACPII-5}$ and Chl $a605_{ACPII-3}$, are observed, which indicates that the outer antenna subunits ACPII-4 and ACPII-5 may transfer energy to the inner antenna ACPII-3 in a relatively low efficiency at the lumenal side (Fig. 9a and Supplementary Table 4).

In the EET from ACPII subunits to the PSII core, five potential EET pathways are identified. At the stromal side, EET mainly takes place through three pathways; one is mediated by ACPII-2, CCPII-S and CP47,

subunits, the other one involves ACPII-6, Unk3 and CP43, and the third one involves ACPII-3 and CP43. In the first pathway, the excitation energy harvested by ACPII-2 may be transferred to CCPII-S, and then further transferred to CP47 of the PSII core. Chl $a602_{CCPII-S}$ is situated between ACPII-2 and CP47, and has Mg-to-Mg distances of 17.18 Å, 21.55 Å and 15.81 Å to Chl $a607_{ACPII-2}$, Chl $a603_{ACPII-2}$ and Chl $a615_{CP47}$ (Fig. 9c). Based on the FRET network, the FRET rate between Chl $a607_{ACPII-2}$ and Chl $a602_{CCPII-S}$, between Chl $a603_{ACPII-2}$ and Chl $a602_{CCPII-S}$, and between Chl $a602_{CCPII-S}$ and Chl $a615_{CP47}$ are 0.221, 0.177 and 1.291 ps$^{-1}$, respectively, enabling efficient EET from ACPII-2 through CCPII-S to CP47 of the PSII core (Fig. 9a and Supplementary Table 4). In the second pathway, the excitation energy harvested by ACPII-6 may be transferred to Unk3, and then further transferred to CP43 of the PSII core. Chl $a602_{ACPII-6}$, Chl $a603_{ACPII-6}$, Chl $a607_{ACPII-6}$, Chl $a101_{Unk3}$, Chl $a507_{CP43}$, and Chl $a513_{CP43}$ form a Chl cluster, facilitating EET from the antenna to PSII core. Chl $a101_{Unk3}$ has Mg-to-Mg distances of 12.65 Å, 13.34 Å, 20.48 Å, 21.03 Å and 15.84 Å to Chl $a603_{ACPII-6}$, Chl $a607_{ACPII-6}$, Chl $a602_{ACPII-6}$, Chl $a507_{CP43}$ and Chl $a513_{CP43}$, respectively (Fig. 9d). Based on the FRET network, the FRET rate between Chl $a603_{ACPII-6}$/Chl $a607_{ACPII-6}$ /Chl $a602_{ACPII-6}$ and Chl $a101_{Unk3}$ are 1.083, 0.421 and 0.185 ps$^{-1}$, respectively, and the Chl pairs between Chl $a101_{Unk3}$ and Chl $a513_{CP43}$/Chl $a507_{CP43}$ have a FRET rate of 0.419 and 0.152 ps$^{-1}$, respectively (Supplementary Table 4), which may result in EET from ACPII-6 through Unk3 to CP43 of the PSII core. In addition, a low FRET rate of 0.049 ps$^{-1}$ was observed between Chl $a607_{ACPII-3}$ and Chl $a506_{CP43}$, which suggests that Chl $a607_{ACPII-3}$ and Chl $a506_{CP43}$ may contribute to energy transfer from ACPII subunits to the PSII core with a lower efficiency compared to the above two pathways at the stromal side (Fig. 9a and Supplementary Table 4).

At the lumenal side, EET pathways from the antennas to the PSII core mainly occur between ACPII-1/ACPII-2 and CP47. The Chl pairs between Chl $a606_{ACPII-1}$ and Chl $a601_{CP47}$, and between Chl $a612_{ACPII-2}$ and Chl $a601_{CP47}$, have FRET rates of 0.099 and 0.041 ps$^{-1}$, respectively, which indicate that the excitation energy harvested by ACPII-1 and ACPII-2 may be transferred to the CP47 of PSII core by these two pathways in a relatively low efficiency (Fig.9a and Supplementary Table 4). This implies that EET in the stromal side may be more efficient than that in the lumenal side. Thus, the CCPII-S and Unk3 subunits previously unidentified in the PSII-ACPII supercomplex may have crucial roles in mediating EET from the outer antennas to the inner PSII core in the cryptophyte algae.

A striking feature of ACPIIs is the presence of cryptophyte-specific composition of carotenoids that are important for enhancing the light-harvesting capacity in the blue green region and photoprotective quenching under excess light illumination. In the current PSII-ACPII structure, some Alxs (Alx 617 and Alx618) exist at the interfaces among ACPII subunits and are surrounded by $a605$ and $a611$ with short distances (<5 Å), allowing them to form Alx-Chl clusters (Fig. 10). This suggests that these Alxs may facilitate either energy transfer or dissipation under strong light conditions. The short distances between Alx615 and $a608$ (3.1 Å) in ACPII-1/2/3/5/6 subunits, between Alx618 and $a614$ (3.2 Å) in ACPII-4 subunit (Supplementary Fig. 10a−f), also imply the possible role of Alxs in the quenching of Chl triplet states. In addition, some of the Cros and Mons are in close proximity to Chls $a$ with the shortest distances <5 Å in ACPII-1/2/4/5/6 subunits (Supplementary Fig. 10g−m). These unusual carotenoids may function as a non-photochemical quenchers to promote energy dissipation from nearby Chls. Thus, this distinct carotenoid composition may be important for the survival of *C. placoidea* inhabiting the marine environment with highly fluctuating light conditions.

## Discussion

Cryptophyte algae are an evolutionarily distinct and ecologically important unicellular eukaryotic algae with diverse photosynthetic pigments. The PSII-ACPII dimer structure solved in this study provides

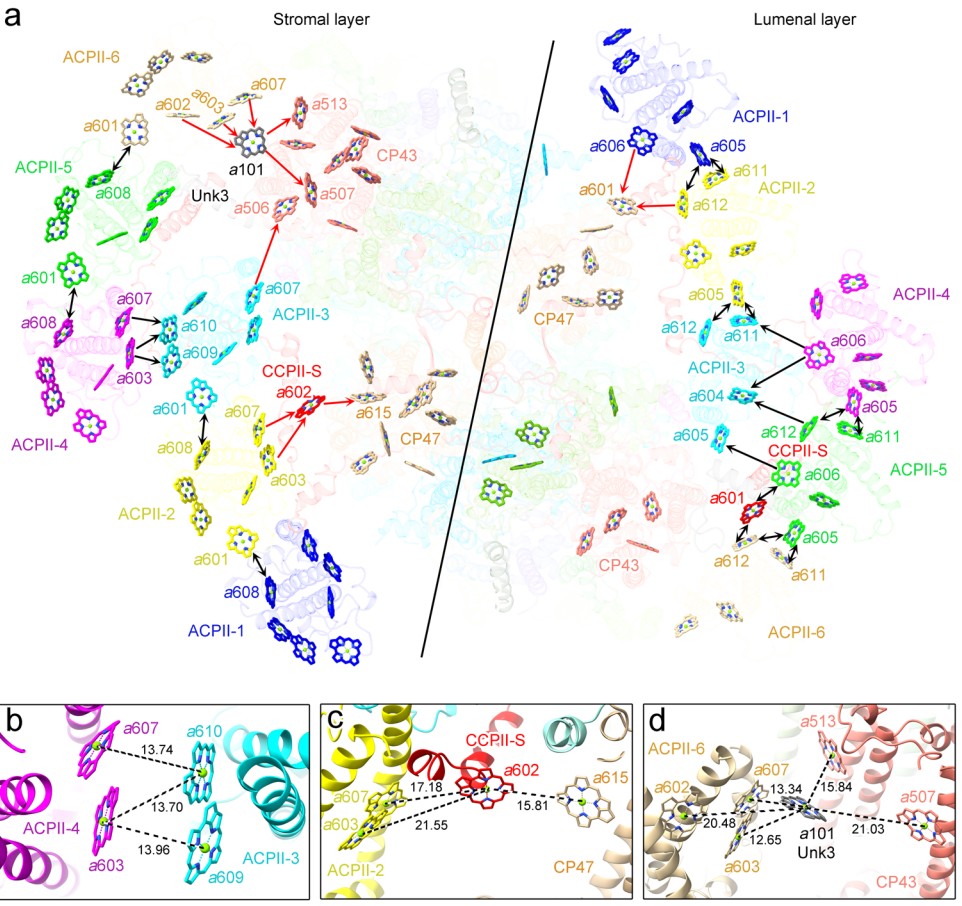

**Fig. 9 | Possible excitation energy transfer pathways in the PSII-ACPII super-complex of *C. placoidea*. a** Distribution of Chls and possible excitation energy transfer pathways in PSII-ACPII supercomplex in the stromal side (left panel) and lumenal side (right panel). Potential energy transfer pathways from ACPIIs to the PSII core are denoted by red arrows, while pathways among ACPIIs are marked with black arrows. **b**–**d** The interfacial pigment between ACPII antennas, PSII core, CCPII-S and Unk3. The pigment interface between ACPII-4 and ACPII-3 (**b**); between ACPII-2, CP47 and CCPII-S (**c**), and between ACPII-6, CP43 and Unk3 (**d**). For clarity, all pigments are colored as that of corresponding subunit, and the phytol chains of the Chl molecules have been omitted.

the atomic structure of a PSII-ACPII supercomplex. This structure consists of two PSII core monomers associated with six symmetrically ACPII proteins on each side, which shows a distinct supramolecular organization in comparison with other PSII-LHCII (FCPII) super-complexes resolved from the green lineage (green algae and higher plants) as well as from the red lineage (diatoms) organisms so far. The existence of peripheral antenna proteins associated with the PSII core in cryptophyte algae has been reported previously, but the exact composition and identities of subunits and pigments were not iden-tified due to the low resolutions previously reported[30]. The current high-resolution cryo-EM density map allows identification of the supramolecular organization of the PSII core and ACPII proteins and the pigment network of the supercomplex. It is considered that the plastids of cryptophyte algae originate from a single secondary endosymbiosis of a red alga in a lineage of Chl-*c* containing algae, and the host cell retains genomes from different organisms[20]. Thus, the structural features of PSII-ACPII revealed in this study are distinct from other organisms and represent the evolutionary changes of different photosynthetic organisms in the course of evolution.

In the PSII core of cryptophyte alga *C. placoidea*, 23 protein sub-units are identified which includes the extrinsic subunits PsbO, PsbV and PsbU, but there are no PsbP and PsbQ found in the green lineage organisms. Among these subunits, 20 subunits are similar to the PSII core from cyanobacteria, red algae and diatoms, and 18 subunits are similar to the PSII core of green algae and plants except the two extrinsic subunits involved in oxygen evolution. This suggests that the major PSII core subunits are highly conserved during evolution, and

the PSII core forms very similar types of dimers in all these organisms. Intriguingly, we identified three subunits Unk1, Unk2, and Unk3 within the PSII core (Fig.1c). Unk1 is located nearby PsbE, PsbF and PsbJ; Unk2 is located in the interface between the two PSII core monomers, and Unk3 is located in the interface between ACPII and the PSII core. Although the detailed structures and functions of these subunits need to be clarified in future studies, it was shown that at least Unk3 may play a crucial role in facilitating the connection between the nearby antenna subunits ACPII-5 and ACPII-6 with the PSII core, and in energy transfer from ACPII to the PSII core.

Regarding the light-harvesting system, one of the distinct features of the PSII-ACPII dimer is the binding of trans-membrane ACPII pro-teins as the major light-harvesting protein, given that *C. placoidea* also contain phycobiliproteins[25–27]. The well-defined cryo-EM density map allowed us to identify the types and structures of all ACPII subunits in the whole PSII-ACPII structure. Detailed comparison between the dif-ferent ACPII subunits showed some obvious differences in both the structure and pigment arrangement of the ACPII subunits. Different from the LHC antennas from other organisms, the *C. placoidea* ACPII antennas contain a distinct composition of carotenoids (Supplemen-tary Fig. 1), but no xanthophyll cycle pigments as reported previously[35,36]. The current study revealed that Alx, a cryptophyte-specific carotenoid with two triple bonds in its structure[31,37,38], is the major carotenoid, and two other unusual carotenoids, Mon and Cro, together with α-carotene, are present as minor carotenoids (Supple-mentary Tables 2, 3). Some of these carotenoids are close to the Chls

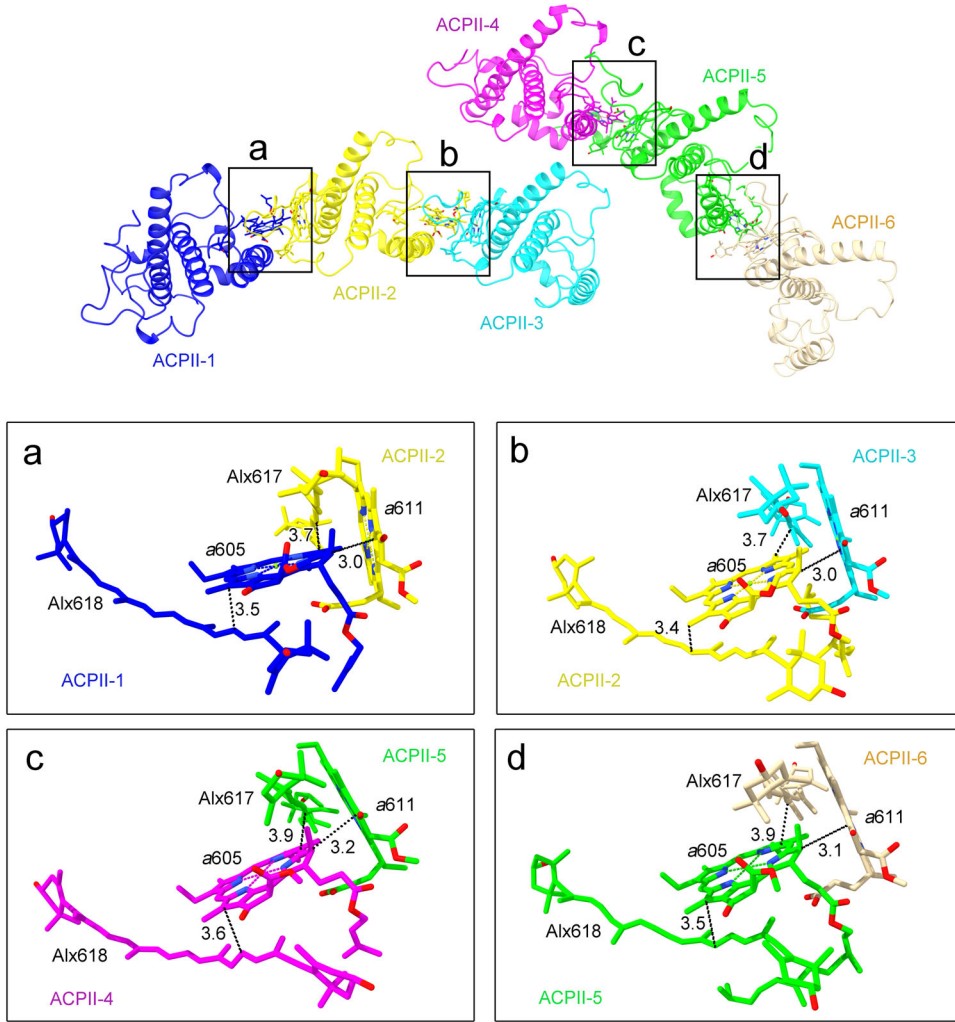

**Fig. 10 | Distribution patterns of typical chlorophylls and alloxanthins at the interface between ACPII subunits that are possibly involved in excitation energy transfer or photoprotection under high light conditions. a–d** Typical Chls and alloxanthins between ACPII-1 and ACPII-2 (**a**), ACPII-2 and ACPII-3 (**b**), ACPII-4 and ACPII-5 (**c**), ACPII-5 and ACPII-6 (**d**). Chls and alloxanthins involved in the possible EETs or photoprotection are depicted in sticks and colored the same as that of the protein subunits. The shortest distances (Å) between two adjacent pigments are labeled in black.

and may be crucial for energy transfer or quenching of the Chl triplet states in high light conditions (Fig. 10), suggesting that the ACPII antennas may play an important role in photoprotection of *C. placoidea* under high light conditions.

Each PSII-ACPII monomer contains six monomeric ACPII proteins acting as a light-harvesting system, which increases the overall antenna cross section to increase the light-harvesting capacity of PSII in *C. placoidea*. The arrangement and assembly of ACPIIs in *C. placoidea* seem to be different from those in diatoms, green algae and higher plants. In the PSII-ACPII structure, six ACPII antennas are divided into two groups, each consisting of 3 adjacent monomers associated in one side of the PSII core. In addition, a special Chl binding subunit named CCPII-S, was observed in PSII-ACPII of *C. placoidea*. This special subunit is not only crucial for the EET from antennas to PSII core, but also involved in extensive interactions with the ACPII antennas and PSII core subunits, mediating the association of ACPII subunits to the PSII core. Together with antenna-PSII core interactions mediated by Unk3, this distinct organization ensures the stable assembly of the PSII-ACPII dimer. It is of note that a special pigment-binding subunit, ACPI-S was also found in the cryptophyte PSI-ACPI supercomplex[32]. This ACPI-S is structurally different from that of CCPII-S of the PSII-ACPII, but functionally similar with that of CCPII-S in mediating the assembly of the supercomplex and energy transfer. Thus, the special Chl *a*-binding

antenna subunit-dependent distinct organization of PSII-ACPII or PSI-ACPI seems to be a common theme in cryptophytes.

Two antenna subunits, ACPII-2 and ACPII-6 in the PSII-ACPII, are found to be identical to the antenna subunits ACPI-2 and ACPI-14 of PSI-ACPI[32]. This indicates that these antenna subunits can be used in both PSI and PSII to regulate energy harvesting and excitation energy distribution between the two photosystems in the cryptophyte algae. This has not been found in other organisms, and may suggest that the cryptophyte algae have not diversified enough in its genes so the same antenna gene has to be used for both photosystems, or it is in a rather preliminary position of evolution that does not allow different genes to be evolved for different photosystems.

ACPII-2/ACPI-2 are located in the inner antenna belts, whereas ACPII-6/ACPI-14 are situated in the outer antenna belts in the PSII-ACPII or PSI-ACPI supercomplexes (Supplementary Fig. 9c, d). Both ACPII-2/ACPI-2 and ACPII-6/ACPI-14 share the same positions in the ACPII and ACPI trimer groups (Supplementary Fig. 9). Further structural comparison shows that ACPII-2 and ACPII-6 related antenna groups of three adjacent ACPII have a high conformational similarity with that of ACPI-2 and ACPI-14 in PSI-ACPI, respectively. This structural similarity may contribute to the coupling/uncoupling of the antennas with the photosystems, resulting in a rearrangement of the antenna proteins in different light conditions. Indeed, a Chl *a/c* antenna dependent state

transition-like mechanism has been observed in the regulation of light harvesting in cryptophyte algae, which may contribute to an efficient carbon fixation in the logarithmic growth phase of the algae[39,40].

In summary, the PSII-ACPII dimer structure of *C. placoidea* offers deep insights into the changes in the PSII core subunits and associated light-harvesting complexes during evolution, as well as excitation-energy transfer mechanisms in the PSII-ACPII supercomplexes of the cryptophyte algae. The existence of different types of ACPII proteins and subunits that have not been observed, and their specific structural features suggest that ACPII antennas are important and essential for the supramolecular organization of PSII as well as for the light harvesting and energy dissipation. The PSII-ACPII dimer has a supramolecular organization and pigment composition distinctly different from those of diatoms, green algae and higher plants, which leads to different pathways of EET in comparison with those of diatoms, green algae and higher plants with trimeric or tetrameric light-harvesting complexes. These results provide an important basis for understanding the regulation of light-energy utilization in cryptophyte algae in deep water where fluctuating light dominates.

## Methods

### Isolation of the PSII-ACPII supercomplex
*Chroomonas placoidea* T11 (a strain generously provided by Prof. Liang Ying, Key Laboratory of Mariculture of Ministry of Education, Ocean University of China, Qingdao, China) was cultured in artificial seawater bubbled with air containing 2–3% $CO_2$ (v/v) at a temperature of 21 °C, under continuous light conditions at a light intensity of 35 to 40 µmol photons $m^{-2}$ $s^{-1}$. Purification of PSII-ACPII supercomplex was performed at 4 °C or on ice under dim green light. Cells grown in the logarithmic phase were collected by centrifugation at 6000 × *g* for 10 min and resuspended in buffer I (30 mM Mes-NaOH, pH 6.5, 1.0 M betaine, 5 mM $MgCl_2$, 5 mM $CaCl_2$) followed by another centrifugation at 6000 × *g* for 10 min. The harvested cells were disrupted by a pressure disruptor (AH-D150) for three cycles at 300 bar, and the unbroken cells were removed by centrifugation at 1000 × *g* for 10 min. Thylakoid membranes were collected by centrifugation at 40,000 × *g* for 30 min and resuspended in buffer II (25 mM Hepes-KOH, pH 7.5, 1.0 M betaine, 5 mM EDTA). The resuspended thylakoid membranes were centrifuged again at 40,000 × *g* for 30 min and resuspended in buffer III (25 mM Hepes-KOH, pH 7.5, 1.0 M betaine, 1 mM $CaCl_2$) to a final concentration of 0.8 mg Chl $ml^{-1}$ and stored in liquid nitrogen until utilization[12].

To isolate the PSII-ACPII supercomplex, thylakoid membranes at a concentration of 0.8 mg Chl $ml^{-1}$ were solubilized with 1.5% (w/v) *n*-dodecyl-α-D-maltopyranoside (α-DDM) (Anatrace) for 20 min on ice, and loaded onto a linear sucrose density gradient (0–1.0 M sucrose) in buffer III containing 0.012% α-DDM, followed by centrifugation at 230,000 × *g* for 18 h (Beckman SW40 rotor). After centrifugation, the PSII-ACPII band was collected and further purified by size-exclusion chromatography (Superose 6 Increase 10/300 GL, Cytiva) in buffer IV (25 mM Hepes-KOH, pH 7.5, 0.6 M betaine, 100 mM NaCl, 1 mM $CaCl_2$, 0.012% α-DDM). The eluted peak was collected and concentrated by an ultrafiltration centrifuge tube (molecular weight cut-off: 100 kDa AMICON, Merck Millipore) to a concentration of 10 mg Chl $ml^{-1}$ and stored in liquid nitrogen. The concentration of Chl *a/c* was determined according to Jeffrey et al.[41].

### Characterization of the PSII–ACPII supercomplex
UV absorption spectra were recorded by a UV-Vis spectrophotometer (UV-2700, Shimadzu, Japan) at room temperature. The fluorescence emission spectra were measured at 77 K with a fluorescence spectrometer (F-4500, Hitachi, Japan) equipped with a xenon lamp, and the spectra were recorded at a wavelength range from 600 to 800 nm with the excitation wavelength of 436 and 460 nm.

Pigment composition was analyzed by high performance liquid chromatography (HPLC) in a Waters e2695 separation module equipped with a Waters 2998 photodiode array detector with the procedures reported previously[15]. The pigments were extracted from the protein sample after size-exclusion chromatography with 90% (v/v) acetone and the resultant extract was injected into a C18 reversed-phase column (Alltima™ C18, 5 µm) pre-equilibrated with solvent A (acetonitrile: water = 9:1). The column was eluted with a linear gradient of 0–100% solvent B (ethyl acetate). The pigments were assigned on the basis of their characteristic absorption spectra and elution times.

The protein composition of the supercomplex was analyzed by SDS-polyacrylamide gel electrophoresis (SDS-PAGE) using a gel containing 16% polyacrylamide and 7.5 M urea[42]. The gels were stained with Coomassie brilliant blue (CBB) R-250. Protein component in one of the CBB-stained bands (Supplementary Fig. 1b) was further analyzed by mass spectrometry with one biological replicate. For this analysis, the CBB-stained band was cut out from the gel and digested using a modified trypsin. The digested peptides were extracted and separated by a home-made, fused silica capillary column (75 µm ID, 150 mm length; Upchurch, Oak Harbor, WA) packed with C18 resin (300 Å, 5 µm; Varian, Lexington, MA). The column was eluted with a 60-minute gradient of acetonitrile at a flow rate of 0.30 µL/min in an EASY-nLC 1000 system interfaced directly with the Thermo Orbitrap Fusion mass spectrometer. The MS/MS spectra from LC-MS/MS run were searched against the specified database using the Proteome Discovery searching algorithm (version 2.5).

Oxygen-evolving activity was determined by a Clark-type oxygen electrode under saturating light at 25 °C in a buffer containing 50 mM Mes-NaOH (pH 6.5), 15 mM $CaCl_2$, 15 mM $MgCl_2$ and 25% glycerol (V/V) at 15 µg Chl/ml. As the electron acceptors, 2,6-dichlorobenzoquinone (DCBQ, 0.5 mM) and potassium ferricyanide (0.5 mM) were used. The oxygen-evolving activity of the purified PSII-ACPII was measured to be 144 µmol $O_2$ (mg Chl)$^{-1}$ $h^{-1}$.

### Sequence analysis of PSII–ACPII
Total RNA was extracted from *C. placoidea* and subjected to transcriptome sequencing by BGI. Sequencing libraries were constructed using DNBSEO-eukaryotic transcriptome/RNA-Seg-PolyA enrichment-BGI kit index. mRNA was enriched and fragmented from the total RNA, followed by reverse transcription using random N6 primers to synthesize double-stranded cDNA. End repair, A-tailing, and adapter ligation were then performed, and the resultant products were amplified with PCR using specific primers. The PCR products were denatured, forming single-stranded DNA, which was then circularized into single-stranded circular DNA libraries using a bridging primer. The libraries were sequenced on an MGISEQ-2000-PE150 + 150 + 10 platform. After sequencing, we filtered out reads of low quality, adapter contamination, and high N content. Then, de novo assembly was performed to obtain the transcriptome sequences.

### Cryo-EM sample preparation and data collection
The concentrated PSII-ACPII sample was diluted to a final concentration of 2 mg Chl *a/c* $ml^{-1}$ using buffer V (25 mM Hepes-KOH, pH 7.5, 1 mM $CaCl_2$, 0.012% α-DDM). Subsequently, 5 µl of the diluted sample was used to load onto a glow-discharged holey carbon grid (CryoMatrix Amorphous alloy film R1.2/1.3, 300 mesh). The grid is then blotted for 4 s with a blot force of 2 using a Vitrobot Mark IV (FEI) at 100% humidity and 4 °C. Cryo-EM images were collected on a Titan Krios microscope (FEI) operated at 300 kV equipped with a Gatan Quantum energy filter (with a slit width of 20 eV) and a K3 camera (Gatan) operated at the super resolution mode, with a magnification of 81,000. Each movie consists of 32 frames with a total dose of ~ 60 e/Å², an exposure time of 1.8 s and a dose rate of 39 e⁻/pixel/s. Data acquisition

was carried out using the EPU software (Thermo Fisher Scientific) with a defocus range of −1.0 to −2.0 µm. The final images were binned, resulting in a pixel size of 1.04 Å for further data processing.

## Cryo-EM image processing

In total 10,663 movies were processed for PSII-ACPII with CryoSPARC[43], from which 1,263,069 particles were automatically selected using crYOLO[44]. After several rounds of selection through 2D classification, 323,227 particles were selected for heterogeneous refinement. A class containing 55.57% particles were selected and further subjected to 2D classification to select templates with fewer particles. These fewer particles were utilized to train a model using the Topaz[45]. This trained model was then employed for particle extraction, yielding 376,092 particles. These particles were subjected to a 2D classification aimed at removing the particles with a lower quality. Subsequently, 326,769 high-quality particles were chosen for a new round of heterogeneous refinement. During this stage, particles making up 74.52% were combined with the previously mentioned 55.57% class, while duplicates were carefully removed. The consolidated set of 305,400 particles was then subjected to non-uniform refinement under a C2 symmetry, which resulted in a structure with a resolution of 2.47 Å based on the criterion of the gold-standard Fourier shell correlation function (GSFSC) = 0.143.

To improve the resolution of the cryo-EM density maps, symmetry expansion and particle subtraction were performed, followed by local refinement targeting the peripheral ACPIIs, which resulted in a final resolution of 2.84 Å for the ACPII part of the supercomplex.

## Model building and refinement

For model building, the structure of *C. gracilis* PSII-FCPII (PDB ID 7VD5)[16] was first manually placed and rigid-body fitted into the 2.47-Å resolution cryo-EM map with UCSF Chimera[46]. The amino acid sequences of D1, D2, CP43, CP47, PsbE, PsbF, PsbH, PsbI, PsbJ, PsbK, PsbL, PsbT, PsbX, PsbY, PsbZ, PsbW, PsbO, PsbU and PsbV subunits were then mutated to the corresponding sequences of *C. placoidea* obtained from the Uniprot database, or the transcriptome sequencing and mass spectrometry performed in the present study. The sequence for the PsbM subunit was not found in the transcriptome analysis, thus we modeled them as the sequence from *Guillardia theta* (strain CCMP2712)[36]. Three subunits Unk1, Unk2, and Unk3 previously unidentified, were modeled as polyalanines, due to the insufficient resolution to identify their amino acid residues. The structures of the six ACPIIs were referenced from the ACPI structure already reported in PSI-ACPI (PDB ID 7Y7B)[32]. Then, the amino acid sequences were mutated to the corresponding sequences of C. placoidea obtained from transcriptome sequencing and mass spectrometry analysis in this study. A complete model was built for the CCPII-S subunit based on the cryo-EM map, transcriptome sequences and mass spectrometry data.

Additional adjustments to the backbone and side chain structures were manually conducted using COOT[47]. The entire PSII-ACPII supercomplex model was refined in real space against the cryo-EM map using Phenix[48]. Models refined by Phenix were then edited in Coot to resolve atomic clashes and geometric issues. The edited model was suffered to another round of refinement using Phenix. These two steps were iteratively performed several times to achieve the final atomic model. The structure was displayed with UCSF Chimera[46] and PyMOL[49].

## Förster resonance energy transfer (FRET) analysis

The FRET rate constants ($k_{FRET}$), defined as $k_{FRET} = (CK^2)/(n^4R^6)$, were calculated according to the FRET theory[50]. In the above equation, $C$ is a factor calculated from the overlap integral between the two Chls, $K$ is the dipole orientation factor, $n$ is the refractive index and $R$ is the distance between two central magnesium atoms of Chls. The $C$ value of 32.26 was applied for Chl $a \rightarrow$ Chl $a$ energy transfer[51], and the $n$ value of

1.55 was taken from ref. 51. $K^2$ was defined as $K^2 = [\hat{u}_D \cdot \hat{u}_A - 3(\hat{u}_D \cdot \hat{R}_{DA})]^2$, where $\hat{u}_D$ and $\hat{u}_A$ are the dipole unit vectors of donor and accepter Chls derived from the vectors of coordinate NB and ND atoms, and $\hat{R}_{DA}$ is the unit vector of the vector from the magnesium of the donor Chl to the magnesium of the acceptor Chl. The FRET rates were computationally calculated using Kim's algorithm available at https://doi.org/10.5281/zenodo.3250649[14,52] on the Python platform (Python v.3.6).

## Reporting summary

Further information on research design is available in the Nature Portfolio Reporting Summary linked to this article.

## Data availability

The cryo-EM density map and atomic model for the PSII-ACPII supercomplex structure have been deposited in the Electron Microscopy Data Bank and the Protein Data Bank with accession codes EMD37414 and 8WB4, and the locally refined cryo-EM map and atomic model of the ACPIIs have been deposited in the Electron Microscopy Data Bank and the Protein Data Bank with accession codes EMD-38419 and 8XKL. The RNA-seq data have been deposited in the NCBI Sequence Read Archive (SRA) database under accession code PRJNA1095808. The data that supports the findings of this study are presented in the paper and/or the Supplementary Information. Source data are provided with this paper.

## Code availability

The python script used for FRET rate calculation is available at https://doi.org/10.5281/zenodo.3250649[14,52].

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

## Acknowledgements

The authors thank W. Tang, L. Wang and Y. Yin of the institute of Botany, CAS for instrumental supports in sample preparation, fluorescence measurement and high-performance liquid chromatography analysis, and X. Meng in Center of Biomedical Analysis, Tsinghua University, for protein MS analysis. The microscope imaging was performed at Beijing National Laboratory for Condensed Matter Physics, Institute of Physics, Chinese Academy of Science and Beijing Branch of Songshan Lake Laboratory for Materials Science. The project was funded by the National Key R&D Program of China (2022YFA0911900, 2022YFC3401800, 2020YFA0907600), the China Postdoctoral Science Foundation (2022M711490), the National Natural Science Foundation of China (32200199), the CAS Project for Young Scientists in Basic Research (YSBR-004), the Strategic Priority Research Program of CAS (XDA26050402), the CAS Interdisciplinary Innovation Team (JCTD-2020-06), the Youth Innovation Promotion Association of CAS (2020081), the Science & Technology Specific Project in Agricultural High-tech Industrial Demonstration Area of the Yellow River Delta (2022SZX12).

## Author contributions

G.H., J.-R.S. conceived the project, Z.M. and G.H. performed the sample preparation and characterization; Z.M. and Xingyue L. carried out cryo-EM grids preparation and the cryo-EM data collection; Xingyue L.

processed the cryo-EM data and reconstructed the cryo-EM map, built the structure model and refined the structure; Z.M., Xingyue L., J.-R.S. and G.H. analyzed the structure; Z.L., L.S., Xiaoyi L., Y.Y., W.W., and T.K. contributed to the discussions and comments on the results; Z.M., Xingyue L., J.-R.S. and G.H. jointly wrote the manuscript.

## Competing interests

The authors declare no competing interests.
