## [Peer Review File · Nature Communications]

Structure and distinct supramolecular organization of a PSII-ACP II dimer from a cryptophyte alga *Chroomonas placoidea*REVIEWER COMMENTS

Reviewer #1 (Remarks to the Author):

The manuscript authored by Mao and co-authors unveils a compelling structure of the Photosystem II supercomplex within a representative of a distinct class of photosynthetic organisms—the cryptophyte algae. The paper, thoughtfully illustrated and meticulously structured, introduces a completely novel perspective. The revelation that RCII forms a dimer, surrounded by a unique light-harvesting antenna featuring chlorophylls a and c alongside various carotenoids, contributes significantly to our understanding. The authors delve into intricate details and novel features of protein-protein and protein-pigment interactions. Notably, their findings highlight how tryptophan coordinates chlorophyll and leucine forms hydrogen bonds with the pigment, while histidine, a typical chlorophyll-coordinating residue, binds to chlorophyll via hydrogen bonding. The manuscript also delves into the binding details of antenna units and antenna-core interactions. A particularly impressive revelation is the close proximity of alloxanthin to certain chlorophylls, suggesting a crucial role for this carotenoid in photoprotection. Overall, this work provides a comprehensive exploration of the Photosystem II supercomplex in cryptophyte algae, offering valuable insights into the intricate interplay of molecular components.

Reviewer #2 (Remarks to the Author):

The manuscript from Mao et al. describes the structure of the PSII-ACPII complex from cryptophyte alga. Given the interesting evolutionary history of cryptophytes and the possible unique NPQ mechanism in this group, the structure is of general interest to the photosynthetic community and merits publication in nature communications. My general impression from the quality of cryoEM data processing is good, although the omission of the focused map is denoted below. The quality of the model refinement is not sufficient for publication, and I have pointed out some of the deficiencies below.

One of the interesting features of this manuscript is the unique organization of ACPII around PSII which is assisted by ACPII-s. This seems to be a common theme in cryptophytes, the authors may want to expand their discussion of this fact.

While I find that the overall quality of writing is good, it's a bit too descriptive in my opinion. For example, in lines 157-162 the authors point to some differences between ACP's but they do not mention any role these differences may play within the PSII-ACPII complex. This repeats itself when ACPII is described and in a few other places in the MS.

Lines 282 – 291 – the authors should not discuss energy transfer as occurring solely from specific chlorophylls. Better terminology would be the “the transfer rate between X and Y is expected to be the fastest based on distance” or similarly (of if the authors choose to calculate rates based on some model

that would be best). The authors should choose their words more carefully when describing these processes.

Lines 305 – 308 – the authors address the possible role of Car and refer to close chl to Car distances as “imply the crucial role of Alxs in energy quenching.”. While this is a possibility, it is impossible to distinguish the role of chl – Car interaction based on structure alone. The authors point to light harvesting and quenching, they should also mention triplet quenching as an important process and be clear with regards to what can be understood from their data.

Line 339 – “it was shown that at least Unk3 not only plays a crucial role in facilitating the connection between the nearby antenna subunits ACPII-5 and ACPII-6 with the PSII core, but also participates in energy transfer from ACPII to the PSII core.” – the authors cannot draw cause and effect conclusions from structural data alone. The phrasing of this sentences hold change.

Lines 342 – 358 – the discussion on NPQ leaves something to be desired for. First and foremost, as far as I know the location of an NPQ mechanism hasn’t been determined in cryptophyte, meaning it is not known if ACPII is even involved in NPQ. The proximity of specific Car to chlorophylls by itself does not say much about their possible involvement in NPQ. As is always the case, the authors should also mention chl triplet quenching as an important mechanism that is known to be affected by this proximity.

While CCP-II-s and APCI-s are clearly different proteins I think the discussion will benefit from the authors commenting on the similarities and differences between them, especially given that some ACP’s seem to be shared between PSI and PSII in in this species. The authors also do not comment on the distribution of APCI2/ACPII2 and APCI14/APCI16 in different species which may be an important point. Since the proteins are identical in sequences the authors should at least present their efforts in trying to identify if they are encoded by the same mRNA or not. If there is no evidence for a different gene coding for these proteins, I don’t see the need for a different name...

Before the acceptant of this manuscript, the refinement of the model should be improved, especially noticeable here is the high sidechain outlier value and the high clash score (above 17, should be below 8 for an acceptable model at the reported resolution).

The authors used a map of ACPII generated using focused refinement to refine the model and identify the different ACP subunits. This map was not supplied as part of the review process and cannot be assessed. The authors should deposit this map together with the ACPII model in the pdb/EMDB before the manuscript is accepted. The images in extended data fig. 3 are not useful to assess the quality of the map and the confidence in side chain assignment, although according to what is possible to see, the map appears to be of good quality.

The occupancy of chains J, U, V, O, Q, G should be assessed by either refinement or classification, the same goes of unk1, 2 and 3 (there may be some overlap between these two groups).

The O₂ evolution activity of this PSII prep is not reported in the methods section, so it’s not clear if this is an active or inactive PSII prep, this must be reported before publication.

Minor changes -

Line 52 – instead of “developed’ I suggest “evolved”.

Lines 122 – 131 – this analysis is important, I suggest that the phrasing of this paragraph should be a bit different to make it more readable. As it stands it’s a bit like reading the supp. information that displays the same data.

Extended Table 1 – “Ratamer outliers” is misspelled.

REVIEWER COMMENTS

Reviewer #1 (Remarks to the Author):

The manuscript authored by Mao and co-authors unveils a compelling structure of the Photosystem II supercomplex within a representative of a distinct class of photosynthetic organisms—the cryptophyte algae. The paper, thoughtfully illustrated and meticulously structured, introduces a completely novel perspective. The revelation that RCII forms a dimer, surrounded by a unique light-harvesting antenna featuring chlorophylls a and c alongside various carotenoids, contributes significantly to our understanding. The authors delve into intricate details and novel features of protein-protein and protein-pigment interactions. Notably, their findings highlight how tryptophan coordinates chlorophyll and leucine forms hydrogen bonds with the pigment, while histidine, a typical chlorophyll-coordinating residue, binds to chlorophyll via hydrogen bonding. The manuscript also delves into the binding details of antenna units and antenna-core interactions. A particularly impressive revelation is the close proximity of alloxanthin to certain chlorophylls, suggesting a crucial role for this carotenoid in photoprotection. Overall, this work provides a comprehensive exploration of the Photosystem II supercomplex in cryptophyte algae, offering valuable insights into the intricate interplay of molecular components.

Author's answers:

We greatly appreciate the reviewer for his/her highly positive evaluation and recommendation of our manuscript.

Reviewer #2 (Remarks to the Author):

The manuscript from Mao et al. describes the structure of the PSII-ACP II complex from cryptophyte alga. Given the interesting evolutionary history of cryptophytes and the possible unique NPQ mechanism in this group, the structure is of general interest to the photosynthetic community and merits publication in nature communications. My general impression from the quality of cryoEM data processing is good, although the omission of the focused map is denoted below. The quality of the model refinement is not sufficient for publication, and I have pointed out some of the deficiencies below.

Author's answers:

We greatly appreciate the reviewer for his/her highly positive and encouraging comments. Regarding the cryo-EM map and the PDB model, we have added the local map and PDB model mentioned in the method section, and also improved the refinement of the PDB file for the whole PSII-ACP II supercomplex. According to the reviewer's comments, we have carefully addressed the issues raised by the reviewer in the revised manuscript, and our point-to-point answers are provided below.

One of the interesting features of this manuscript is the unique organization of ACP II around PSII which is assisted by ACP II-s. This seems to be a common theme in cryptophytes, the authors may want to expand their discussion of this fact.

Author's answers:

We thank the reviewer for this important comment. In this study, a special Chl a-binding antenna subunit (CCP II-S) not observed in any other previously characterized PSII-LHC II complexes, was identified, which contributes to the association and possible energy transfer between the ACP IIs and PSII core in each PSII-ACP II monomer. This is functionally similar with the ACP I-S, a pigment-binding subunit that is only found in cryptophyte PSI complex, which mediates the association and energy transfer between the outer and inner ACP Is in cryptophyte PSI (Zhao et al., Plant Cell, 2023, 35, 2449-2463). As the reviewer pointed out, the assembly of PSII with ACP II or PSI with ACP Is seems to be dependent on the unique Chl a-binding antenna subunit, and may be a common feature in cryptophytes. We added these points into the discussion section in the revised manuscript as follows, according to the reviewer's suggestions.

“It is of note that a special pigment-binding subunit, ACP I-S was also found in the cryptophyte PSI-ACP I supercomplex³². This ACP I-S is structurally different from that of CCP II-S of the PSII-ACP II, but functionally similar with that of CCP II-S in mediating the assembly of the supercomplex and energy transfer. Thus, the special Chl a-binding antenna subunit-dependent unique organization of PSII-ACP II or PSI-ACP I seems to be a common theme in cryptophytes.” (Lines 369-374, revised manuscript).

While I find that the overall quality of writing is good, it's a bit too descriptive in my opinion. For example, in lines 157-162 the authors point to some differences between ACP's but they do not mention any role these differences may play within the PSII-ACPII complex. This repeats itself when ACPII is described and in a few other places in the MS.

Author's answers:

We agree with the reviewer's comments that there are some places that are too descriptive but no mentions are made regarding the roles they may have. We have deleted the repetitive description in lines 145-146, 265, and 344-346 of the original manuscript. A sentence is added to the revised manuscript regarding the possible role of the difference between ACPIIs as follows:

"These structural differences may be a result of the specific position that each antenna subunit occupies, thus contributing to the unique arrangement and stable assembly of the ACPIIs in the PSII-ACPII supercomplex." (Lines 156-158, revised manuscript)

We also modified some other places to reflect the possible roles of the unique ACPIIs subunits in the cryptophyte PSII-ACPII supercomplex.

Lines 282 – 291 – the authors should not discuss energy transfer as occurring solely from specific chlorophylls. Better terminology would be the "the transfer rate between X and Y is expected to be the fastest based on distance" or similarly (of if the authors choose to calculate rates based on some model that would be best). The authors should choose their words more carefully when describing these processes.

Author's answers:

We thank the reviewer for this important comment. The energy transfer pathways are related not only to the inter-pigment distances, but also with the relative orientation of the pigments. In this study, we discuss the possible excitation energy transfer pathways among Chls mainly based on their distances, as the effects of relative orientations of the chlorin rings of Chls require more extensive and quantitative calculation. According to the reviewer's suggestions, we have rephrased the sentences regarding the energy transfer processes, and added the following sentence at the start of description of the energy transfer, to reflect the situation more accurately in the revised manuscript.

"It should be pointed out that the following descriptions are based on pigment-pigment distances only, and the EET pathways also depend on the orientations of pigments involved, which should be treated explicitly in future studies." (Lines 270-272, revised manuscript)

Lines 305 – 308 – the authors address the possible role of Car and refer to close chl to Car distances as “imply the crucial role of Alxs in energy quenching.”. While this is a possibility, it is impossible to distinguish the role of chl – Car interaction based on structure alone. The authors point to light harvesting and quenching, they should also mention triplet quenching as an important process and be clear with regards to what can be understood from their data.

Author’s answers:

We agree with the reviewer’s comments that it is impossible to distinguish the role of Chl – Car interaction based on structure alone. According to the reviewer’s suggestions, we have rephrased the expression about the possible role of Car, and added the triplet quenching as an important process.

Line 339 – “it was shown that at least Unk3 not only plays a crucial role in facilitating the connection between the nearby antenna subunits ACPII-5 and ACPII-6 with the PSII core, but also participates in energy transfer from ACPII to the PSII core.” – the authors cannot draw cause and effect conclusions from structural data alone. The phrasing of this sentences hold change.

Author’s answers:

We fully agree with the reviewer’s comments and have rephrased the sentence as “it was shown that at least Unk3 may play a crucial role in facilitating the connection between the nearby antenna subunits ACPII-5 and ACPII-6 with the PSII core, and energy transfer from ACPII to the PSII core.” (Lines343-345, revised manuscript), according to the reviewer’s suggestions.

Lines 342 – 358 – the discussion on NPQ leaves something to be desired for. First and foremost, as far as I know the location of an NPQ mechanism hasn’t been determined in cryptophyte, meaning it is not known if ACPII is even involved in NPQ. The proximity of specific Car to chlorophylls by itself does not say much about their possible involvement in NPQ. As is always the case, the authors should also mention chl triplet quenching as an important mechanism that is known to be affected by this proximity.

Author’s answers:

We thank the reviewer for this important comment. The review is right that the exact location of an NPQ mechanism hasn’t been determined in cryptophyte and the proximity of specific Car to chlorophylls by itself does not say much about their possible involvement in NPQ. According to the reviewer’s comments and suggestions, we have modified the expression and mentioned Chl triplet quenching as an important, alternative mechanism in the discussion of the Car function.

While CCP-II-s and APCI-s are clearly different proteins I think the discussion will benefit from the authors commenting on the similarities and differences between them, especially given that some ACP's seem to be shared between PSI and PSII in in this species. The authors also do not comment on the distribution of APCI2/ACP-II2 and APCI4/ACP-II6 in different species which may be an important point. Since the proteins are identical in sequences the authors should at least present their efforts in trying to identify if they are encoded by the same mRNA or not. If there is no evidence for a different gene coding for these proteins, I don't see the need for a different name...

Author's answers:

Thanks for your comments. For the CCP-II-s and APCI-s, based on the structure, they are clearly different proteins from PSII and PSI, respectively. We have added corresponding discussions (Lines 369-374) on the similarities and differences between them in the revised manuscript. Regarding APCI-2/ACP-II-2 and APCI-14/ACP-II-6 seem to be shared between PSI and PSII in in this species, we have added additional discussion (Lines 378-381) in the revised manuscript. According to the reviewer's comments, we have added the comments on the distribution of APCI2/ACP-II2 and APCI4/ACP-II6 in different photosystems (Lines 382-385) in the revised manuscript. Although our transcriptome sequencing suggests they are encoded by the same mRNA, there is no strong evidence to prove they are encoded by different or same genes for these proteins due to the lack of the genomic sequences of this species. The different names given here are just for clarity to indicate it is one of the components of PSII-ACP-II or PSI-ACP-I supercomplexes.

Before the acceptant of this manuscript, the refinement of the model should be improved, especially noticeable here is the high sidechain outlier value and the high clash score (above 17, should be below 8 for an acceptable model at the reported resolution).

Author's answers:

We thank the reviewer for pointing out this. According to the reviewer's comments, we have improved the quality of the refinement of the model. The sidechain outlier value and the high clash score are optimized to an acceptable value of 0.08% and 7.06, respectively. The new model has been re-deposited to the Protein Data Bank (PDB) and the validation information was updated in the Supplementary Table 1 in the revised manuscript.

The authors used a map of ACP-II generated using focused refinement to refine the model and identify the different ACP subunits. This map was not supplied as part of the review process and cannot be assessed. The authors should deposit this map together with the ACP-II model in the pdb/EMDB before the manuscript is accepted. The images in extended data fig. 3 are not useful to assess the quality of the map and the confidence in side chain assignment, although according to what is possible to see, the map appears to be of good quality.

Author's answers:

Thank you. According to the reviewer's comments, we have deposited the local map and corresponding PDB model of ACPII at a resolution of 2.84 Å into the Electron Microscopy Data Bank (EMDB) and the Protein Data Bank (PDB) with ID codes of EMD 38419 and PDB 8XKL, respectively. We also supplemented the local map and model alongside with the manuscript. This will facilitate the assessment of the quality of our model and the confidence in side-chain assignment.

The occupancy of chains J, U, V, O, Q, G should be assessed by either refinement or classification, the same goes of unk1, 2 and 3 (there may be some overlap between these two groups).

Author's answers:

Thank you. According to the reviewer's suggestion, we assessed the occupancy of chains J, U, V, O, Q, G as well as the unidentified subunits unk1, unk2, and unk3 by 3D classification. We reprocessed the final dataset consisting of 305,400 particles with a focused 3D classification, specifically targeting the region of extrinsic subunits. This resulted in ten types of distinct maps as listed in the following figure, and the presence or absence of these chains are summarized in the following table.

4	×	×	×	×	√	√	√
5	×	×	×	×	√	√	√
6	×	×	×	×	×	√	√
7	×	×	×	×	×	√	√
8	×	×	×	×	×	×	√
9	√	√	√	√	√	√	×
10	×	×	×	×	×	×	×

The O₂ evolution activity of this PSII prep is not reported in the methods section, so it's not clear if this is an active or inactive PSII prep, this must be reported before publication.

Author's answers:

We thank the reviewer for pointing out this. According to the reviewer's suggestion, we have measured the O₂ evolution activity of the PSII preparation used for cryo-EM analysis and the detailed information about the O₂ evolution characterization has been added in the Methods section (Lines 456-460) in the revised manuscript. The measurement indicated that this PSII prep has an O₂ evolution activity of ~144 μmol O₂ (mg Chl)⁻¹ h⁻¹, which is relatively low due to the loss of the extrinsic proteins in a large fraction of the sample (see the above occupancy analysis).

Minor changes

Line 52 – instead of “developed” I suggest “evolved”.

Author's answers:

Thank you. According to the reviewer's suggestion, we have replaced the “developed” with “evolved” in the revised manuscript.

Lines 122 – 131 – this analysis is important, I suggest that the phrasing of this paragraph should be a bit different to make it more readable. As it stands it's a bit like reading the supp. information that displays the same data.

Author's answers:

Thank you. According to the reviewer's suggestion, we have rephrased this paragraph as follows in the revised manuscript.

“Structural comparison of the PSII cores reveals that the root-mean-square deviation (RMSD) values of α-carbon atoms between the PSII core of cryptophyte algae and that

of a cyanobacterium⁴, a red alga³⁴, a green alga¹⁴, a diatom¹⁶, and a higher plant¹⁸ (including only the same subunits in these PSII cores) are 0.827, 0.910, 1.035, 0.743, and 0.983, respectively (Supplementary Fig. 4a-e).” (Lines 121-125, revised manuscript).

Extended Table 1 – “Ratamer outliers” is misspelled.

Author’s answers:

Thank you. We have corrected the typo “Ratamer outliers” to “Rotamer outliers” in the revised manuscript.

REVIEWERS' COMMENTS

Reviewer #1 (Remarks to the Author):

I recommend to publish this version.

Reviewer #3 (Remarks to the Author):

Reviewer's comment:

The evaluation of models of the PSII-ACPII and ACPII supercomplexes demonstrates a very good alignment with the cryo-EM maps, underscoring their accuracy. Improved validation statistics, such as root mean square deviations of bond lengths and angles, clashscore, and percentages of rotamer and Ramachandran outliers at (Supplementary Table 1), fall within the range typical for high-quality models in this field. These measures, particularly the low root mean square deviations, highlight the precise atomic positioning in the models, while the low clashscore and outlier percentages indicate the structural integrity and reliability of the model conformations.

In addition, I have a comment on the presented analysis of the excitation energy transfer (EET). The current EET analysis primarily considers the distances between chlorophyll molecules. However, this approach might not always accurately identify the predominant pathways for energy transfer. It is essential to incorporate the orientation factor in EET calculations, as the spatial orientation of chlorophylls significantly influences the efficiency of energy transfer. The proximity of chlorophyll molecules does not necessarily guarantee effective energy transfer. To facilitate a more comprehensive analysis, I recommend utilizing the Python script available at <https://doi.org/10.5281/zenodo.3250649>, as detailed in Sheng X. et al. (2019) in their study, 'Structural insight into light harvesting for photosystem II in green algae' published in *Nature Plants* 5: 1320-1330. This script includes the orientation factor in its calculations, enabling a more accurate analysis and understanding of EET pathways in photosynthetic systems.

REVIEWERS' COMMENTS

Reviewer #1 (Remarks to the Author):

I recommend to publish this version.

Author's answers:

We thank the reviewer for his/her recommendation of our manuscript.

Reviewer #3 (Remarks to the Author):

Reviewer's comment:

The evaluation of models of the PSII-ACP II and ACP II supercomplexes demonstrates a very good alignment with the cryo-EM maps, underscoring their accuracy. Improved validation statistics, such as root mean square deviations of bond lengths and angles, clashscore, and percentages of rotamer and Ramachandran outliers at (Supplementary Table 1), fall within the range typical for high-quality models in this field. These measures, particularly the low root mean square deviations, highlight the precise atomic positioning in the models, while the low clashscore and outlier percentages indicate the structural integrity and reliability of the model conformations.

Author's answers:

We greatly appreciate the reviewer for his/her highly positive evaluation of our improvement of the models of the PSII-ACP II and ACP II supercomplexes.

In addition, I have a comment on the presented analysis of the excitation energy transfer (EET). The current EET analysis primarily considers the distances between chlorophyll molecules. However, this approach might not always accurately identify the predominant pathways for energy transfer. It is essential to incorporate the orientation factor in EET calculations, as the spatial orientation of chlorophylls significantly influences the efficiency of energy transfer. The proximity of chlorophyll molecules does not necessarily guarantee effective energy transfer. To facilitate a more comprehensive analysis, I recommend utilizing the Python script available at <https://doi.org/10.5281/zenodo.3250649>, as detailed in Sheng X. et al. (2019) in their study, 'Structural insight into light harvesting for photosystem II in green algae' published in Nature Plants 5: 1320-1330. This script includes the orientation factor in its calculations, enabling a more accurate analysis and understanding of EET pathways in photosynthetic systems.

We fully agree with the reviewer's comments that the energy transfer pathways are related not only to the inter-pigment distances, but also with the relative orientation of the pigments and it is essential to incorporate the orientation factor in EET calculations. According to the reviewer's suggestions, we have analyzed the pathways of energy transfer by the calculations of FRET network using the Python script available at

<https://doi.org/10.5281/zenodo.3250649>, as suggested by the reviewer. The corresponding results are described in the revised manuscript (lines 274-331, Page 11-13), and related results is updated as Figs. 8, 9 in the revised manuscript.